

**Evaluating residual error approaches for post-processing monthly**
**and seasonal streamflow forecasts**
Fitsum Woldemeskel[1], David McInerney[2],  Julien Lerat[3], Mark Thyer[2], Dmitri Kavetski[2,4],
Daehyok Shin[1], Narendra Tuteja[3] and George Kuczera[4]
(1) Bureau of Meteorology, VIC, Australia
(2) School of Civil, Environmental and Mining Engineering, University of Adelaide, SA, Australia
(3) Bureau of Meteorology, ACT, Australia
(4) School of Engineering, University of Newcastle, Callaghan, NSW, Australia
Correspondence email: fitsum.woldemeskel@bom.gov.au



## Abstract

Streamflow forecasting is prone to substantial uncertainty due to errors in meteorological forecasts,
hydrological model structure and parameterization, as well as in the observed rainfall and streamflow
data used to calibrate the models. Statistical streamflow post-processing is an important technique
available to improve the probabilistic properties of the forecasts. This study evaluates three residual error
models based on the logarithmic (Log), log-sinh (Log-Sinh) and Box-Cox with $\lambda = 0.2$ (BC0.2)
transformation schemes and identifies the best performing scheme for post-processing monthly and
seasonal (3-months) streamflow forecasts, such as those produced by the Australian Bureau of
Meteorology. Using the Bureau's operational dynamic streamflow forecasting system, we carry out
comprehensive analysis of the three post-processing schemes across 300 Australian catchments with a
wide range of hydro-climatic conditions. Forecast verification is assessed using reliability and sharpness
metrics, as well as the Continuous Ranked Probability Skill Score (CRPSS). Results show that the
uncorrected forecasts (i.e. without post-processing) are unreliable at half of the catchments. Post-
processing using the three residual error models substantially improves reliability, with more than 90%
of forecasts classified as reliable. In terms of sharpness, the BC0.2 scheme significantly outperforms the
Log and Log-Sinh schemes. Overall, the BC0.2 scheme achieves reliable and sharper-than-climatology
forecasts at a larger number of catchments than the Log and Log-Sinh error models. This study is
significant because the reliable and sharper forecasts obtained using the BC0.2 post-processing scheme
will help water managers and users of the forecasting service to make better-informed decisions in
planning and management of water resources.
**Keywords**: seasonal streamflow forecasts, residual error models, post-processing, Box-Cox
transformation









**Key points**

1. Uncorrected and post-processed streamflow forecasts (using three residual error models, based
on the Log, Log-Sinh and BC0.2 transformations respectively) are evaluated over 300 diverse
Australian catchments.
2. Post-processing enhances streamflow forecast reliability, increasing the percentage of sites with
reliable predictions from 50% to over 90%.
3. The BC0.2 transformation achieves significantly better forecast sharpness than the Log-sinh and
Log transformations, particularly in dry catchments.


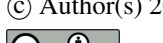



## 1   Introduction

Hydrological forecasts provide crucial supporting information on a range of water resource management
decisions, including (depending on the forecast lead-time) flood emergency response, water allocation
for various uses, and drought risk management (Li et al., 2016; Turner et al., 2017). The forecasts,
however, should be thoroughly verified and proved to be of sufficient quality to support decision-making
and to meaningfully benefit the economy, environment and society.
Sub-seasonal and seasonal streamflow forecasting systems can be broadly classified into two types
(Crochemore et al., 2016):
*i. Dynamic modelling systems*. Here, a hydrological model is commonly developed at a daily time-step
to capture key hydrological processes. The model is calibrated against observed streamflow using
historical rainfall and potential evaporation data. Once the model is calibrated, rainfall forecasts from a
numerical climate model are used as an input to produce daily streamflow forecasts, which are then
aggregated to the time scale of interest and post-processed using statistical models. Examples of
operational services based on the dynamic approach include the Australian Bureau of Meteorology's
dynamic modelling system (Laugesen et al., 2011; Tuteja et al., 2011; Lerat et al., 2015); the
Hydrological Ensemble Forecast Service (HEFS) of the US National Weather Service (NWS) (Brown
et al., 2014; Demargne et al., 2014); the Hydrological Outlook UK (HOUK) (Prudhomme et al., 2017);
and the short-term forecasting European Flood Alert System (EFAS) (Cloke et al., 2013).
*ii. Statistical modelling systems.* Here, a statistical model based on relevant predictors is applied directly
at the time scale of interest. A number of predictors have been considered in the literature, including
antecedent rainfall and streamflow, soil moisture, depth and extent of snow cover, and climate indices
derived from sea surface temperature (Robertson and Wang, 2009, 2011; Wang et al., 2009; Tang and
Lettenmaier, 2010; Lü et al., 2016; Zhao et al., 2016). The Bureau of Meteorology's Bayesian Joint
Probability (BJP) forecasting system is an example of an operational service based on a statistical
approach (Senlin et al., 2017).
Hybrid systems that share some characteristics of dynamic and statistical approaches have also been
investigated. For example, Robertson et al. (2013) and Humphrey et al. (2016) used dynamic model
simulations as predictors for statistical models.
Dynamic and statistical approaches have distinct advantages and limitations. Dynamic systems can
potentially provide realistic responses in unfamiliar climate situations as it is possible to impose physical
constraints in such situations (Wood and Schaake, 2008). In comparison, statistical models have the
flexibility to include features that may lead to more reliable predictions. For example, the BJP model



uses climate indices (e.g. NINO3.4), which are typically not used in dynamic approaches. That said, the suitability of statistical models for the analysis of non-stationary catchment and climate conditions is questionable (Wood and Schaake, 2008).

Streamflow forecasts built on hydrological models are affected by uncertainty in a number of factors, including rainfall forecasts, observed rainfall and streamflow data, as well as the parametric and structural uncertainty of the hydrological model. Progress has been made towards reducing biases and characterizing the sources of uncertainty in streamflow forecasting. These advances include improving rainfall forecasts through post-processing ( Robertson et al., 2013b; Crochemore et al., 2016), accounting for input, parametric and/or structural uncertainty (Kavetski et al., 2006; Kuczera et al., 2006; Renard et al., 2011; Tyralla and Schumann, 2016) and using data assimilation techniques (Dechant and Moradkhani, 2011). Although these steps may improve some aspects of the forecasting system, a residual bias may nonetheless remain. Such bias can only be reduced via post-processing, which, if successful, will improve forecast accuracy and reliability (Madadgar et al., 2014; Lerat et al., 2015).

This study focuses on improving streamflow forecasting using dynamic approaches, by identifying residual error models suitable for post-processing hydrological forecasts at monthly and seasonal time-scales. A number of post-processing approaches have been investigated in the literature, including quantile mapping (Hashino et al., 2007), Bayesian frameworks (Pokhrel et al., 2013; Robertson et al., 2013a), as well as methods based on state-space models and wavelet transformations (Bogner and Kalas, 2008). Wood and Schaake (2008) used the correlation between forecast ensemble means and observations to generate a conditional forecast. Compared with the traditional approach of correcting individual forecast ensembles, the correlation approach improved forecast skill and reliability. In another study, Pokhrel et al. (2013) implemented a Bayesian Joint Probability (BJP) method to correct biases, update predictions and quantify uncertainty in monthly hydrological model predictions in 18 Australian catchments. The study found that the accuracy and reliability of forecasts improved. More recently, Mendoza et al. (2017) evaluated a number of seasonal streamflow forecasting approaches, including purely statistical, purely dynamical, and hybrid approaches. Based on analysis of catchments contributing to five reservoirs, the study concluded that incorporating catchment and climate information into post-processing improves forecast skill. While the above review mainly focused on post-processing at sub-seasonal and seasonal forecasts (as it is the main focus of the current study), post-processing is also commonly applied to short-range forecasts (e.g. Li et al., 2016; Seo et al., 2006) and to long-range forecasts up to 12 months ahead (Bennett et al., 2016).

In most streamflow post-processing approaches, a residual error model is applied to quantify forecast uncertainty. Most residual error models are based on least squares techniques with weights and/or data



transformations (e.g. Carpenter and Georgakakos, 2001; Li et al., 2016; Seo et al., 2006). In order to
produce post-processed streamflow forecasts, a daily-scale residual error model is used in the calibration
of hydrological model parameters, and a monthly/seasonal-scale residual error model used as part of
streamflow post-processing to quantify the forecast uncertainty. In a recent study, McInerney et al.
(2017) concluded that residual error models based on Box-Cox transformations with fixed parameter
values are particularly effective for daily scale predictions, yielding significant improvements in dry
catchments. While McInerney et al. (2017) used observed rainfall to force the hydrological model, and
evaluated daily streamflow predictions, this study investigates whether these findings generalize to
monthly and seasonal forecasts using forecast rainfall.
An important aspect of this work is its focus on general findings applicable over diverse hydro-
climatological conditions. Most of the studies in the published literature use a limited number of
catchments and case studies to test prospective methods. Dry catchments, characterised by intermittent
flows and frequent low flows, pose the greatest challenge to hydrological models (Ye et al., 1997;
Knoche et al., 2014). Yet the provision of good quality forecasts across a large number of sites is an
essential attribute of national scale operational forecasting service, especially in large countries with
diverse climatic and catchment conditions, such as Australia.
This paper aims to develop streamflow post-processing approaches suitable for use in an operational
streamflow forecasting service. More specifically, our aims are:
**Aim 1**: Evaluate the value of streamflow forecast post-processing by comparing forecasts with no post-
processing (hereafter called 'uncorrected' forecasts) against post-processed forecasts.
**Aim 2**: Evaluate three residual error models proposed in recent publications, namely the Log, Box-Cox
(McInerney et al., 2017) and Log-Sinh (Wang et al., 2012) schemes, for monthly and seasonal
streamflow post-processing.
**Aim 3**: Evaluate the generality of results over a diverse range of hydro-climatic conditions, in order to
ensure the recommendations are robust in the context of an operational streamflow forecasting service.
To achieve these aims, we use the operational monthly and seasonal (3-months) dynamic streamflow
forecasting system of the Australian Bureau of Meteorology (Lerat et al., 2015). We evaluate the residual
error models across 300 catchments across Australia, with detailed analysis of dry and wet catchments.
Forecast verification is carried out using Continuous Ranked Probability Skill Score (CRPSS) as well
as metrics measuring reliability and sharpness, which are important aspects of a probabilistic forecast
(Wilks, 2011). These metrics are used by the Bureau of Meteorology to describe streamflow forecast
performance of the operational service.



The rest of the paper is organised as follows. The forecasting methodology is described in Section 2 and
application studies are described in Section 3. Results are presented in Section 4, followed by discussions
and conclusions in Sections 5 and 6 respectively.

## 2  Seasonal Streamflow forecasting methodology

### 2.1  Overview

The streamflow forecasting system adopted in this study is based on the Bureau of Meteorology's
dynamic modelling system (Figure 1). This dynamic modelling system uses daily rainfall forecasts as
inputs into a daily rainfall-runoff model to produce daily streamflow forecasts. These streamflow
forecasts are then aggregated in time and post-processed to produce monthly and seasonal streamflow
forecasts, which are issued each month. In general, two steps are involved: simulation and forecasting.

### 2.2  Simulation Step

In the simulation step, the daily rainfall-runoff model is calibrated to observed daily streamflow using
observed rainfall (Jeffrey et al., 2001) as forcing.
The rainfall-runoff model GR4J (Perrin et al., 2003) is used as it has been proven to provide (on average)
good performance across a large number of catchments ranging from semi-arid to temperate and tropical
humid (Perrin et al., 2003; Tuteja et al., 2011). The calibration of the hydrological model is based on the
weighted least squares likelihood function, similar to that outlined in Evin et al. (2014). Markov Chain
Monte Carlo (MCMC) analysis is used to estimate posterior parametric uncertainty (Tuteja et al., 2011).
Following MCMC analysis, 40 random sets of GR4J parameters are retained and used in the forecast
step.

### 2.3  Forecast Step

Once the hydrological model is calibrated, daily downscaled rainfall forecast from the Bureau of
Meteorology's global climate model, namely the Predictive Ocean Atmosphere Model for Australia
POAMA-2 (Hudson et al., 2013), are routed through the hydrological model to produce daily
uncorrected streamflow forecasts. The atmospheric component of POAMA-2 uses a spatial scale of
approximately $250 \times 250$ km (Charles et al., 2013). To estimate catchment-scale rainfall, a statistical
downscaling model based on an analogue approach (Timbal and McAvaney, 2001) was applied. In the
analogue approach, local climate information is obtained by matching analogous previous situations to
the predicted climate. To this end, an ensemble of 166 rainfall forecast time series (33 POAMA
ensembles $\times$ 5 replicates from downscaling + 1 ensemble mean) were generated. These forecasts are
then input into GR4J and propagated using the 40 GR4J parameter sets to obtain 6640 ($166 \times 40$) daily
streamflow forecasts. The daily streamflow forecasts generated using GR4J are then aggregated to



monthly and seasonal time scales to produce ensembles of 6640 uncorrected monthly and seasonal
forecasts.

## 2.4 Streamflow post-processing

Post-processing of streamflow forecasts is intended to remove systemic biases in the mean, variability
and persistence of the uncorrected forecasts, which arise due to inaccuracies in the downscaled rainfall
forecasts (e.g. errors in downscaled forecast rainfall from approximately a 250 km grid to the catchment
scale) and in the hydrological model (e.g. due to the effects of data errors on the model calibration and
due to structural errors in the model itself).
The streamflow post-processing method used in this work consists of fitting a statistical model to the
streamflow forecast residual errors, defined by the differences between the observed and forecast
streamflow time series over a calibration period. Typically these residual errors are heteroscedastic and
exhibit persistence. Heteroscedasticity is handled using data transformations (e.g. the Box-Cox
transformation), whereas persistence is represented using autoregressive models (e.g., the lag-one
autoregressive model, AR(1)). We begin by describing the two major steps of the streamflow post-
processing procedure (Sections 2.4.1 and 2.4.2), and then describe the transformations under
consideration (Section 2.5).

### 2.4.1 Calibration of residual error model parameters

The parameters of the streamflow post-processing model are calibrated in the following three steps:
*Step 1*: Compute the transformed forecast residuals for month or season $t$ of the calibration period:

$$\eta_t = Z(\widetilde{Q_t}) - Z(Q_t^F) \tag{1}$$

where $\eta_t$ is the normalised residual, $\widetilde{Q_t}$ is the observed streamflow, $Q_t^F$ is the median of the uncorrected
streamflow forecast ensemble, and $Z$ is a transformation function used to reduce the heteroscedasticity
and skewness of the residuals (Wang et al., 2012; McInerney et al., 2017). The data transformation
functions are detailed in Section 2.5.
*Step 2*: Compute the standardised residuals according to:

$$\nu_t = (\eta_t - \mu_\eta^{m(t)}) / \sigma_\eta^{m(t)} \tag{2}$$

where $\mu_\eta^{m(t)}$ and $\sigma_\eta^{m(t)}$ are the monthly mean and standard deviation of the residuals in the calibration
period for the month $m(t)$. The standardisation process in equation (2) aims to account for seasonal
variations in the distribution of residuals.





The quantities $\mu_\eta^{m(t)}$ and $\sigma_\eta^{m(t)}$ are calculated independently as the sample mean and standard deviation of
residuals for each monthly period (for a monthly forecast) or three-monthly period (for seasonal
forecasts). The standardised residuals $v_t$ are assumed to have a zero mean and unit standard deviation.
*Step 3*: Assume the standardised residuals are described by a first order autoregressive (AR(1)) model:

$$v_{t+1} = \rho v_t + y_{t+1} \qquad\qquad (3)$$

where $\rho$ is the AR(1) coefficient and $y_{t+1} \sim N(0, \sigma_y)$ is the innovation assumed to follow a Gaussian
distribution.
The parameters $\rho$ and $\sigma_y$ are estimated based on the method of moments: $\rho$ is set to the sample auto-
correlation of the standardized residuals $\mathbf{v}$, and $\sigma_y$ is set to the sample standard deviation of the
observed innovations $\mathbf{y}$, which are calculated from the standardized residuals $\mathbf{v}$ by re-arranging
equation (3).

### 237    2.4.2  Streamflow forecasting

Once the streamflow post-processor has been calibrated, the post-processed streamflow forecasts for a
given period are computed. For a given ensemble member *j*, the following steps are applied (note the
additional subscript $j$ for the ensemble number):
*Step 1*: Sample the innovation $y_{t+1,j} \leftarrow N(0, \sigma_y)$.
*Step 2*: Generate the standardized residuals $v_{t+1,j}$ using equation (3). Here $v_{t,j}$ is determined using
equation (2) and $\eta_{t,j}$ using equation (1), which uses the streamflow forecasts and observations from the
previous time step *t*.
*Step 3*: Compute the normalized residuals $\eta_{t+1,j}$ by "de-standardizing" $v_{t+1,j}$:

$$\eta_{t+1,j} = \sigma_\eta^{m(t)} v_{t+1,j} + \mu_\eta^{m(t)} \qquad\qquad (4)$$

*Step 4*: Back-transform each normalized residual $\eta_{t+1,j}$ to obtain the post-processed streamflow forecast:

$$Q^{PP}_{t+1,j} = Z^{-1}[Z(Q^F_{t+1}) + \eta_{t+1,j}] \qquad\qquad (5)$$

Steps 1-4 are repeated for all ensemble members (6640 in our case).
Note that the above algorithm may occasionally generate negative streamflow, which is then set to zero.
This aspect is discussed in Section 5.6.





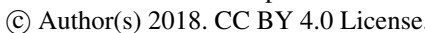

## 2.5 Transformations used in the residual error model
The observed streamflow and median streamflow forecast are transformed in Step 1 of streamflow post-
processing (Section 2.4.1), to account for the heteroscedasticity and skewness of the forecast residuals.
To achieve Aim 2 of this study, we trial three different transformations, namely the logarithmic, log-
sinh and Box-Cox transformations.
### 2.5.1 Logarithmic (Log) transformation
The logarithmic (Log) transformation is
$$Z(Q) = \log(Q + c) \tag{6}$$

The offset $c$ ensures the transformed flows are defined when $Q = 0$. Here we set $c = 0.01 \times (\tilde{Q})_{ave}$
, where $(\tilde{Q})_{ave}$ is the average observed streamflow over the calibration period. The use of a small fixed
value for $c$ is common in the literature for coping with zero flow events (Wang et al., 2012).
### 2.5.2 Log-Sinh transformation
The Log-Sinh transformation (Wang et al., 2012) is
$$Z(Q) = \frac{1}{b} \log \left[ \sinh(a + bQ) \right] \tag{7}$$

The parameters $a$ and $b$ are calibrated for each month by maximising the p-value of the Shapiro-Wilk
test (Shapiro and Wilk, 1965) for normality of the residuals, $v$. This pragmatic approach is part of the
existing Bureau's operational dynamic streamflow forecasting system (Lerat et al., 2015).
### 2.5.3 Box-Cox
The Box-Cox transformation (Box and Cox, 1964) is
$$Z(Q; \lambda, c) = \frac{(Q + c)^{\lambda} - 1}{\lambda} \tag{8}$$

where $\lambda$ is a power parameter and $c = 0.01 \times (\tilde{Q})_{ave}$. Following the recommendations of McInerney et
al. (2017), the parameter $\lambda$ is fixed to 0.2. This avoids the need to calibrate $\lambda$, and related problems with
doing so.
### 2.5.4 Rationale for selecting transformational approaches
The Log transformation is a widely used transformation that is simple to implement; McInerney et al.
(2017) reported that in daily scale modelling it produced the best reliability in perennial catchments
(from a set of eight residual error schemes, including standard least squares, weighted least squares, BC,



Log-Sinh and reciprocal transformation). However, the Log transformation performed poorly in
ephemeral catchments, where its precision was far worse than in perennial ones.
The Log-Sinh transformation is an alternative to the Log and BC transformations proposed by Wang et
al. (2012) to improve the precision at higher flows. The Log-Sinh approach has been extensively applied
to water forecasting problems (see for example, Del Giudice et al., 2013; Robertson et al., 2013b, Bennett
et al., 2016). However, McInerney et al. (2017) found that in daily scale modelling of perennial
catchments, when using observed rainfall, the Log-Sinh scheme did not improve on the Log
transformation (its parameters tend to calibrate to values for which the Log-Sinh transformation reduces
to the Log transformation).
Finally, the BC transformation with fixed $\lambda = 0.2$ is recommended by McInerney et al. (2017) as one of
only two schemes (from the set of eight, see above) that achieve "Pareto-optimal" (e.g., Cohon and
Marks, 1975) performance in terms of reliability, precision and bias, across both perennial and
ephemeral catchments. McInerney et al. (2017) also found that calibrating $\lambda$ did not generally improve
predictive performance, due to the inferred value being dominated by the fit to the low flows at the
expense of the high flows.

### 2.6 Summary

In the remainder of the paper, the term "uncorrected forecasts" refers to streamflow forecasts obtained
using steps in Sections 2.1-2.3, and the term "post-processed forecasts" refers to forecasts based on a
streamflow post-processing model, which includes the standardization and AR(1) model from Section
2.4, as well as a transformation (Log, Log-Sinh or BC0.2) from Section 2.5. As the streamflow residual
error models considered in this work differ solely in the transformation used, they will be referred to as
the Log, Log-Sinh and BC0.2 schemes.

## 3 Application

### 3.1 Data

A comprehensive set of 300 catchments representative of the diverse Australian hydro-climatic
conditions is used, with locations shown in Figure 2. In each catchment, data from 1980-2008 is used.
Observed daily rainfall data was obtained from the Australian Water Availability Project (AWAP)
(Jeffrey et al., 2001). Potential evaporation and observed streamflow data were obtained from the Bureau
of Meteorology. Rainfall forecasts from POAMA-2 were downscaled based on an analogue approach
(Timbal and McAvaney, 2001). These 300 sites are currently being evaluated as part of the expansion
of dynamic modelling for the seasonal streamflow forecasting service of the Bureau of Meteorology.
The figure also shows the Koppen climate zones.



### 3.2    Catchment classification

The performance of the residual error models is evaluated separately in dry versus wet catchments. In this work, the classification of catchments into dry and wet is based on the aridity index (AI) according to the following equation

$$AI = \frac{P}{PET} \tag{9}$$

where P is the total rainfall volume and PET is the total potential evapotranspiration volume. The aridity index has been used extensively to identify drought and wetness conditions of hydrological regimes ( Zhang et al., 2009; Carrillo et al., 2011; Sawicz et al., 2014).

Catchments with $AI < 0.5$ are categorised as "dry", which corresponds to hyper-arid, arid and semi-arid classifications suggested by the United Nations Environment Programme (Middleton et al., 1997). Conversely, catchments with $AI \geq 0.5$ are classified as "wet". Overall, about 28% of catchments used in this work are classified as dry.

### 3.3    Cross-validation procedure

The forecast verification is carried out using a moving-window cross-validation framework, as shown in Figure 3. Suppose we are validating the streamflow forecasts in year $j$ ($j = 1990$ in Figure 3). In this case the calibration is carried out using all years except $j$, $j+1$, $j+2$, $j+3$ and $j+4$. The four-year period after year $j$ are excluded to avoid the effects of memory in the hydrological model. The process is then repeated for each year during 1980-2008. Once the validation has been carried out for each year, the results are concatenated together to produce a single "validation" time series, for which the verification metrics are calculated.

### 3.4    Verification metrics

The goal of the forecasting exercise is to maximise sharpness without sacrificing reliability (Gneiting et al., 2005; Wilks, 2011; Bourdin et al., 2014). Therefore the performance of uncorrected and post-processed streamflow forecasts is evaluated using reliability and sharpness metrics, as well as the Continuous Ranked Probability Skill Score (CRPSS, see section 3.4.3). Note that the Bureau of Meteorology uses Root Mean Squared Error (RMSE) and Root Mean Squared Error in Probability (RMSEP) scores in the operational service in addition to CRPSS, however, RMSE and RMSEP results have not been included in the current paper.

Forecast verification metrics are computed separately for each forecast month. To facilitate the comparison and evaluation of streamflow forecast performance in different streamflow regimes, the high





and low flow months are defined using long-term average streamflow data calculated for each month –
"high flow" months are the 6 months with the highest average streamflow, while low flows are the 6
months with the lowest average streamflow. Note that although the verification metrics are computed
for each month separately, indices denoting the month are excluded from Equations (10), (11) and (12)
below to avoid cluttering the notation.

### 3.4.1  Reliability

The reliability of forecasts is evaluated using the Probability Integral Transform (PIT) (Dawid, 1984;
Laio and Tamea, 2007). To evaluate and compare reliability across 300 catchments, the p-value of the
Kolmogorov-Smirnov (KS) test applied to the PIT is used. In this study, forecasts with PIT plots where
the KS test yields a p-value $\geq 5\%$ are classified as "reliable".

### 3.4.2  Sharpness

The sharpness of forecasts is evaluated using the ratio of inter-quantile ranges (IQR) of streamflow
forecasts and a historical reference (Tuteja et al., 2016). The following definition is used:

$$IQR_q = \frac{1}{n}\sum_{i=1}^{n} \frac{F_i(100-q) - F_i(q)}{C_i(100-q) - C_i(q)} \times 100 \ \% \tag{10}$$

where $IQR_q$ is the $IQR$ value corresponding to percentile $q$, $F_i(q)$, and $C_i(q)$ are the $q$th percentiles of
forecast and historical reference for years $i = 1, 2, ..., N$, respectively.
An $IQR_q$ of 100% indicates a forecast with the same sharpness as the reference, an $IQR_q$ below 100%
indicates forecasts that are sharper (predictive limits that are smaller) than the reference, and an $IQR_q$
above 100% indicates forecasts that are less sharp (predictive limits are wider) than the reference. We
consider $IQR_{99}$, i.e., the $IQR$ at the 99 percentile, in order to detect forecasts with unreasonably long
tails in their predictive distributions.

### 3.4.3  CRPS skill score (CRPSS)

The $CRPS$ metric quantifies the difference between a forecast distribution and observations, as follows
(Hersbach, 2000):

$$CRPS = \int_{-\infty}^{\infty}\left[F_f(y) - F_o(y)\right]^2 dy \tag{11}$$

where $F_f$ and $F_o$ are the cumulative distribution functions (cdfs) of the streamflow forecast and
observation, respectively. The cdf of the observation is taken as the Heaviside step function at the
observed point value.



The $CRPS$ summarises the reliability, sharpness and bias attributes of the forecast (Hersbach, 2000). A
"perfect" forecast – namely a point prediction that matches the actual value of the predicted quantity –
has $CRPS^P = 0$. In this work, we use $CRPS$ skill score, CRPSS, defined by:
$$CRPSS = \frac{CRPS^F - CRPS^C}{CRPS^P - CRPS^C} \times 100\%$$    (12)
where $CRPS^F$, $CRPS^C$ and $CRPS^P$ represent the $CRPS$ value for model forecast, climatology and
"perfect" forecast respectively. A higher CRPSS indicates better performance, with a value of 0
representing the same performance as climatology.

### 3.4.4 Historical reference

The IQR and CRPSS metrics are defined as skill scores relative to a reference forecast. In this work, we
use the climatology as the reference forecast, as it represents the long-term climate condition. To
construct these "climatological forecasts", we used the same historical reference as the operational
seasonal streamflow forecasting service of the Bureau of Meteorology. This reference is resampled from
a Gaussian probability distribution fitted to the observed streamflow data transformed using the log-sinh
transformation (Equation 7). This approach leads to more stable and continuous historical reference
estimates than sampling directly from the empirical distribution of historical streamflow, and can be
computed at any percentile (which facilitates comparison with forecast percentiles). Although the choice
of a particular reference affects the computation of skill scores, it does not affect the ranking of error
models when the same reference is used, which is the main aim of this paper.

### 3.4.5 Summary Skill: Summarising forecast performance using multiple metrics

When evaluating forecast performance, a focus on any single individual metric can lead to misleading
interpretations. For example, two forecasts might have a similar sharpness, however, if one is not
reliable, then it can over or underestimate risk which could lead to a sub-optimal decision by forecast
users (e.g. a water resources manager).
Given inevitable trade-offs between individual metrics (McInerney et al., 2017), it is important to
consider multiple metrics jointly rather than individually. Following the approach suggested by Gneiting
et al. (2007), we consider a forecast to have "high skill" when it is both reliable and has a better sharpness
than climatology. To determine the "summary skill" of the forecasts in each catchment, we evaluate the
total number of months (out of 12) in which forecasts are reliable (i.e., with a p-value greater than 5%)
and sharper than the climatology (i.e., IQR99 < 100%). Accordingly, a catchment is classified as having
high (low) summary skill if it has a 10-12 months (0-2 months) with reliable forecasts that are shaper





than climatology. Note that we do not include the CRPSS in the summary skill, because the CRPSS does
not provide an independent measure of forecast attribute (see Section 3.4.3 for more details).
A table providing the percentage of catchments with high and low summary skills is used to summarise
forecasts performance. In addition, to identify any geographic trends in the forecast performance, the
summary skills are plotted on a map. The summary skills together with individual skill score values are
used to evaluate the overall forecast performance.

## 4 Results

Results for monthly and seasonal streamflow forecasts are now presented. Section 4.1 compares the
uncorrected and post-processed streamflow forecast performance. Section 4.2 evaluates the performance
of post-processed streamflow forecasts obtained using the Log, Log-Sinh and BC0.2 schemes. The
CRPSS, reliability and sharpness metrics are presented in Figure 4 and Figure 5 for monthly and seasonal
forecasts respectively.
Initial inspection of results found considerable overlap in the performance metrics achieved by the error
models. To determine whether the differences in metrics are consistent over multiple catchments, the
Log and Log-Sinh schemes are compared to the BC0.2 scheme. This comparison is presented in
Figure 6 and Figure 7 for monthly and seasonal forecasts respectively. The BC0.2 scheme is taken as
the baseline because inspection of Figure 4 and Figure 5 suggests that the BC0.2 scheme has better
median sharpness than the Log and Log-Sinh schemes, over all the catchments and for both high and
low flow months individually.
The streamflow forecast time-series and corresponding skill for a single representative site, Dieckmans
Bridge, are presented in Figures 8 and 9, respectively.
The results are presented separately for wet and dry catchments, as well as separately for high and low
flow months (Sections 3.2 and 3.4). The summary skills of the monthly and seasonal forecasts are
presented in Figure 10 and Figure 11. The figures include a histogram of summary skills across all
catchments to enable comparison between the uncorrected and the post-processing approaches.

### 4.1 Comparison of uncorrected and post-processed streamflow forecasts: Individual metrics

In terms of CRPSS, largest improvement as a result of post-processing using the Log, Log-Sinh and
BC0.2 schemes occurs in dry catchments for both monthly (Figure 4c) and seasonal forecasts (Figure
5c). For example, when post-processing is used with the three transformation schemes, the median
CRPSS of monthly forecasts in dry catchments increases from approximately 7% (high flow months)



and -15% (low flow months) to more than 10% (Figure 4c) for both high and low flows. Visible
improvement is also observed in dry catchments for seasonal forecasts, however, the improvement is
not as pronounced as for monthly forecasts (Figure 5c).
In terms of reliability, the performance of uncorrected streamflow forecasts is poor, with about 50% of
the catchments being characterized by unreliable forecasts at both the monthly and seasonal time scales
(Figure 4 and Figure 5, middle row). In comparison, post-processing using the three transformation
approaches produces much better reliability, achieving reliable forecasts in more than 90% of the
catchments.
In terms of sharpness, the uncorrected forecasts and the BC0.2 post-processed forecasts are generally
sharper than forecasts generated using the other transformations (Figures 4g and 5g). The use of post-
processing achieves much better sharpness than uncorrected forecasts for low flow months, particularly
in dry catchments. For example, for low flow months in dry catchments (Figure 4i), the median IQR99
is greater than 200% while similar values range between 40-100% for post-processed forecasts.
Similarly, for seasonal forecasts, post-processing approaches improve the median sharpness from in
excess of 150% (uncorrected forecasts) to 50%-110% (Figure 45i).
**4.2    Comparison of residual error models for post-processing: Individual metrics**
In terms of CRPSS, Figure 4 (a, b, c) and Figure 5 (a, b, c) show considerable overlap in the boxplots
corresponding to all three residual error models, both in wet and dry catchments. This finding suggests
little difference in the performance of the residual error models, and is further confirmed by Figure 6 (a,
b, c) and Figure 7 (a, b, c), which show boxplots of the differences between the CRPSS of the Log and
Log-Sinh schemes versus the CRPSS of the BC0.2 scheme. Across all catchments, the distribution of
these differences is approximately symmetric with a mean close to 0. In dry catchments, the BC0.2
slightly outperforms the Log scheme for high flow months and the Log-Sinh scheme slightly
outperforms the Log scheme for low flow months. Overall, these results suggest that none of the Log,
Log-Sinh or BC0.2 schemes is consistently better in terms of CRPSS values.
In terms of reliability, post-processing using any of the three residual error models produces reliable
forecasts at both monthly and seasonal scales, and in the majority of the catchments (Figure 4 and Figure
5, middle row). The median p-value is approximately 60% for monthly forecasts compared with 45%
for seasonal forecasts. This indicates that better reliability is achieved at shorter lead times. Median
reliability is somewhat reduced when using the BC0.2 scheme compared to the Log and Log-Sinh
schemes in wet catchments (Figure 6e), but not so much in dry catchments (Figure 8f). Nevertheless,
the monthly and seasonal forecasts are reliable in 96% and 91% of the catchments, respectively. The



corresponding percentages for the Log scheme are 97% and 94%, and for Log-Sinh they are 95% and
90%.

In terms of sharpness, the BC0.2 scheme produces much sharper forecasts than the Log and Log-Sinh
schemes. This finding holds in all cases (i.e., high/low flow months and wet/dry catchments), both for
monthly and seasonal forecasts (Figure 4 and Figure 5, bottom row). The plot of differences in the
sharpness metric (Figure 6 and Figure 7, bottom row) clearly highlights this improvement. In half of the
catchments, during both high and low flow months, the BC0.2 scheme improves the IQR99 by 30% or
more compared to the Log and Log-Sinh schemes. In dry catchments, the magnitude of the
improvements are higher than wet catchments. For example, in dry catchments during high flow months,
the BC0.2 scheme improves on the IQR99 of Log and Log-Sinh by 40-60% in over a half of the
catchments, and by as much as ~170%-190% in a quarter of the catchments.
To highlight the implication of these results, a representative streamflow forecast time-series at
Dieckmans Bridge catchment (site id: 145010A) is shown in Figure 8 and performance metrics
calculated over six high and low flow months are shown in Figure 9. In terms of reliability, the
uncorrected forecast has a number of observed data points outside the 99% predictive range (Figure 8a).
This is an indication that the forecast is unreliable. This finding can also be confirmed from the
corresponding p-value in Figure 9, which shows that the forecast is below the reliability threshold during
most of the high flow months and also during some low flow months. In terms of sharpness, Log and
Log-Sinh schemes produce a wider 99% predictive range than BC0.2 (Figures 8 and 9).

### 4.3    Comparison of summary skill between uncorrected and post-processing approaches

Figure 10 and Figure 11 show the geographic distribution of the summary skill of the uncorrected and
post-processing approaches for monthly and seasonal forecasts respectively. The summary skill
aggregates multiple verifications metrics: it represents the number of months with streamflow forecasts
that are both reliable and exhibit a sharpness that is better than climatology. Table 1 provides a summary
of the percentage of catchments with high and low summary skill for the uncorrected and post-processing
approaches for monthly and seasonal forecasts. Catchments with high (low) summary skill are defined
as those with 10-12 months (0-2 months) with forecasts that are reliable and sharper than climatology.
At the monthly scale (Figure 10 and Table 1), we obtain the following key findings:
•   Uncorrected forecasts perform worse than post-processing techniques in the sense that they have
low summary skill in the largest percentage of catchments (16%). The percentage of catchments
where high summary skill is achieved is 40%.





- Post-processing forecasts with the Log and Log Sinh scheme, reduces the percentage of catchments with low summary skills to 2% and 7% respectively. However, the percentage of catchments with high summary skill also decreases (in comparison to unprocessed forecasts), to 33% for both Log and Log-Sinh.

- Post-processing with the BC0.2 scheme provides the best performance, with the smallest percentage of catchments with low summary skills (<1%) and the largest percentage of catchments with high summary skills (84%). Figure 10 shows the improvement achieved by the BC0.2 scheme (compared to the Log/Log Sinh schemes) is most pronounced in NSW and in the tropical catchments in QLD and NT. The few catchments where the BC0.2 scheme does not achieve a high summary skill are located in the north and north-west of Australia.

The findings for seasonal forecasts (Figure 11 and Table 1) are as follows:

- Log scheme has the largest percentage (19%) of catchments with low summary skill and a relatively small percentage of catchments (9%) with high summary skill (9%).

- Post-processing forecasts with the Log and Log-Sinh schemes reduces the percentages of catchments with low summary skill to 18% and 17% respectively. The percentage of catchments with high summary skill increases to 12% and 22% respectively.

- Post-processing with the BC0.2 scheme once again provides a clear improvement: it produces forecasts with low summary skill in only 2% of the catchments, and achieves high summary skill in 54% of the catchments. Figure 11, shows that similar to monthly forecasts, the biggest improvements occur in the NSW and Queensland regions of Australia.

Overall, the summary skills of post-processing approaches are lower for seasonal forecasts than for monthly forecasts. Table 1 shows that, across all schemes, BC0.2 results in a larger percentage of catchments with low summary skill and a larger percentage of catchments with high summary skill.

## 4.4  Summary

Sections 4.1-4.3 show that post-processing produces major improvements in reliability, as well as CRPSS and sharpness, particularly in dry catchments. Although all three residual error models under consideration provide improvements in some of the performance metrics, the BC0.2 scheme consistently produces better sharpness than the Log and Log-Sinh schemes, while maintaining similar reliability and CRPSS. This finding holds for both monthly and, to a less degree, seasonal forecasts. Of the three residual error models, the BC0.2 scheme improves by the largest margin the percentage of sites and the number of months where the post-processed forecasts are reliable and sharper than climatology.



## 5   Discussion

### 5.1   Benefit of post-processing

A comparison of uncorrected and post-processed streamflow forecasts was provided in Section 4.1. Uncorrected forecasts have reasonable sharpness (except for dry catchments), but suffer from low reliability: uncorrected forecasts are unreliable at approximately 50% of the sites. In wet catchments, poor reliability is due to overconfident forecasts, which appears a common concern in dynamic forecasting approaches (Wood and Schaake, 2008). In dry catchments, uncorrected forecasts are both unreliable and exhibit poor sharpness. Post-processing is thus particularly important to correct for these shortcomings and improve forecast skill. In this study, all post-processing models provide a clear improvement in reliability and sharpness, especially in dry catchments. The value of post-processing is more significant in dry catchments than in wet catchments (Figure 4 and Figure 5). This finding can be attributed to the challenge of capturing key physical processes in modelling dry and ephemeral catchments (Ye et al., 1997) as well as the challenge of achieving accurate rainfall forecasts in arid areas. In such cases, the hydrological model forecasts are particularly poor and leave a lot of room for improvement: post-processing can hence make a big difference on the quality of results.

### 5.2   Interpretation of differences between residual error models

We now discuss the large differences in sharpness between the BC0.2 scheme versus the Log and Log-Sinh schemes. The Log-Sinh residual error model was designed by Wang et al. (2012) in order to improve the reliability and sharpness of predictions, particularly for high flows, and has worked well when used as part of statistical modelling system for operational streamflow forecasts by the Bureau of Meteorology. The Log-Sinh transformation corresponds to a variance stabilizing function that (for certain parameter values) tapers off for high flows. In theory, this feature can prevent the explosive growth of predictions for high flows that can occur with the log and Box-Cox residual error models (especially when $\lambda < 0$).

McInerney et al. (2017) found that, when modelling perennial catchments at the daily scale, the Log-Sinh scheme did not achieve better sharpness than the Log scheme; instead, the parameters for the Log scheme tended to converge to values for which the tapering off of the Log-Sinh scheme occurs well outside the range of simulated flows, and hence the Log-Sinh scheme effectively reduces to the Log scheme. In contrast, the Box-Cox error model when using a fixed $\lambda > 0$ has a variance-stabilizing function that gradually flattens as streamflow increases, i.e., it exhibits the "desired" tapering-off behaviour.





Our findings in this study confirm the insights of McInerney et al. (2017) – namely that the Log-Sinh
scheme produces comparable sharpness to the Log scheme – across a larger number of catchments. This
finding indicates that insights from modelling residual errors at the daily scale apply at least to some
extent to streamflow forecast post-processing at the monthly and seasonal scales. Note the minor
difference in the treatment of the offset parameter $c$ in equation (6): in the Log scheme used in McInerney
et al. (2017) this parameter is inferred, whereas in this study it is fixed a priori. This minor difference
does not impact on the qualitative behaviour of the error models, as described earlier in this section. The
BC0.2 scheme provides an opportunity to further improve forecast performance relative to what is
currently possible using the Log and Log-Sinh schemes when used as residual error post-processor of
forecasts in a dynamical modelling systems.
**5.3   Importance of using multiple metrics to assess forecast performance**
The study results show that relying on a single metric for evaluating forecast performance can lead to
sub-optimal conclusions. For example, if one considers the CRPSS metric alone, all post-processing
schemes yield comparable performance and there is no basis for favouring any single one of them.
However, once sharpness is taken into consideration explicitly, the BC0.2 scheme can be recommended
due to significantly better sharpness than the Log and Log-Sinh schemes. Similarly, comparisons based
solely on CRPSS might suggest reasonable performance of the uncorrected forecasts (55%-80% of
months have CRPSS > 0 depending on high/low flow months and monthly/seasonal forecasts), yet once
reliability is considered explicitly, it is found that uncorrected forecasts are unreliable at approximately
50% of the catchments. Note that, for example, CRPSS reflects an implicitly weighted combination of
reliability, sharpness and bias characteristics of the forecasts (Hersbach, 2000), whereas the reliability
and sharpness metrics are specifically designed to target reliability and sharpness attributes respectively.
These findings highlight the value of multiple independent performance metrics and diagnostics that
evaluate specific attributes of the forecasts, and highlight important limitations of aggregate measures
of performance (Clark et al., 2011).
A number of challenges and questions remain in regards to selecting the verification metrics for specific
forecasting systems and applications. An important question is how to include user needs into a forecast
verification protocol. This could be accomplished by tailoring the evaluation metrics to the requirements
of users. Another key question is to what extent do measures of forecast skill correlate to the economic
and/or social value of the forecast? This question was investigated by Murphy and Ehrendorfer (1987)
and Wandishin and Brooks (2002), who found the relationship between quality and value of a forecast
to be essentially nonlinear: an increase in forecast quality may not necessarily lead to a proportional
increase in its value.





### 5.4 Importance of performance evaluation over large numbers of catchments

When designing an operational forecast service for locations with streamflow regimes as diverse and variable as in Australia (Taschetto and England, 2009), it is essential to thoroughly evaluate multiple modelling methods over multiple locations to ensure the findings are sufficiently robust and general. This was the major reason for considering the large set of 300 catchments in our study. This setup also yields valuable insights into spatial patterns in forecast performance. For example, the Log and Log-Sinh schemes perform relatively well in catchments in South-Eastern Australia, and relatively worse in catchments in Northern and North-Eastern Australia (Figure 10 and Figure 11). In contrast, the BC0.2 scheme performs well across the majority of the catchments in all regions included in the evaluation. The evaluation over a large number of catchments in different hydro-climatic regions is clearly beneficial to establish the robustness of post-processing methods. Restricting the analysis to a smaller number of catchments would have led to less conclusive findings.

### 5.5 Implication of results for water resource management

The management of water resources, for example, deciding which water source to use for a particular purpose or allocating environmental flows, requires an understanding of the current and future availability of water. For water resources systems with long hydrological records, water managers have devised techniques to evaluate current water availability, water demand and losses. However, one of the main unknowns is the volume of future system inflows. Streamflow forecasts thus provide crucial information to water managers and users regarding the future availability of water, thus helping reduce uncertainty in decision making. The ability of the BC0.2 post-processing scheme to improve forecast sharpness (precision) while maintaining forecast accuracy and reliability can hence lead to improved operational planning and management of water resources.

### 5.6 Treatment of zero flows

The post-processing approach using the three residual error models described above does not make special provision for zero flows in the calibration approach. Robust handling of zero flows in statistical models is an active research area (Wang and Robertson, 2011; Smith et al., 2015), and advances in this area are certainly relevant to seasonal streamflow forecasting.

## 6 Conclusions

This study focused on developing robust streamflow forecast post-processing schemes for an operational forecasting service at the monthly and seasonal time scales. For such forecasts to be useful to water managers and decision-makers, they should be reliable and exhibit sharpness that is better than climatology.



We investigated streamflow forecast postprocessor schemes employing residual error models based on
three data transformations, namely the logarithmic (Log), log-sinh (Log-Sinh) and Box-Cox
transformation with $\lambda = 0.2$ (BC0.2). The Australian Bureau of Meteorology's dynamic modelling
system was used as the platform for the empirical analysis, which was carried out over 300 Australian
catchments with diverse hydro-climatic conditions.
The outcomes of this study are:
1. Uncorrected forecasts (no post-processing) perform poorly in terms of reliability, which is an
indication that forecast uncertainties are misrepresented. All three post-processing schemes
substantially improve the reliability of streamflow forecasts, both in terms of the dedicated
reliability metric and in terms of the summary skill given by the CRPSS;
2. From the post-processing schemes considered in this work, the BC0.2 scheme is found best
suited for operational application. The BC0.2 scheme provides the sharpest forecasts without
sacrificing reliability, as measured by the reliability and CRPSS metrics. In particular, the BC0.2
scheme produces forecasts that are both reliable and sharper than climatology at substantially
more sites than the alternative Log and Log-Sinh schemes.
In conclusion, this study developed a robust streamflow forecast post-processing scheme that achieves
reliable and consistently sharper-than-climatology streamflow forecasts. This scheme is well suited for
operational application, and offers the opportunity to improve decision support, especially at sites where
climatology is presently used to guide operational decisions.
## 7  Data Availability
The data underlying this research can be accessed from the following links: Observed rainfall data
(http://www.bom.gov.au/climate/); POAMA rainfall forecast (http://poama.bom.gov.au/); and observed
streamflow data (http://www.bom.gov.au/waterdata/).
## 8  Acknowledgments
Data for this study is provided by the Australian Bureau of Meteorology. This work was supported by
the Australian Research Council grant LP140100978 with the Australian Bureau of Meteorology and
South East Queensland Water.




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








Table 1. Percentage of catchments with high and low summary skill for the different residual error
schemes for both monthly and seasonal forecasts. High (low) summary skill is defined as the percentage
of catchments with 10-12 months (0-2 months) reliable forecasts that are sharper than climatology.

| Residual Error Scheme | Uncorrected forecasts | Log | Log-Sinh | BC0.2 |
|---|---|---|---|---|
| *Monthly Forecasts* | | | | |
| High Summary Skill | 40% | 33% | 33% | 84% |
| Low Summary Skill | 16% | 2% | 7% | <1% |
| *Seasonal Forecasts* | | | | |
| High Summary Skill | 46% | 9% | 20% | 54% |
| Low Summary Skill | 14% | 19% | 17% | 2% |









**Figures**

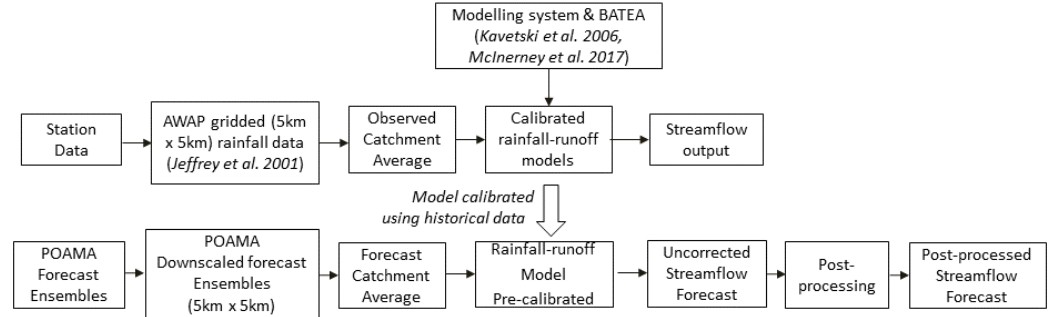

Figure 1: Schematic of the dynamic streamflow forecasting system used in this study. A similar approach is used by the Australian Bureau of Meteorology for its monthly and seasonal streamflow forecasting service.








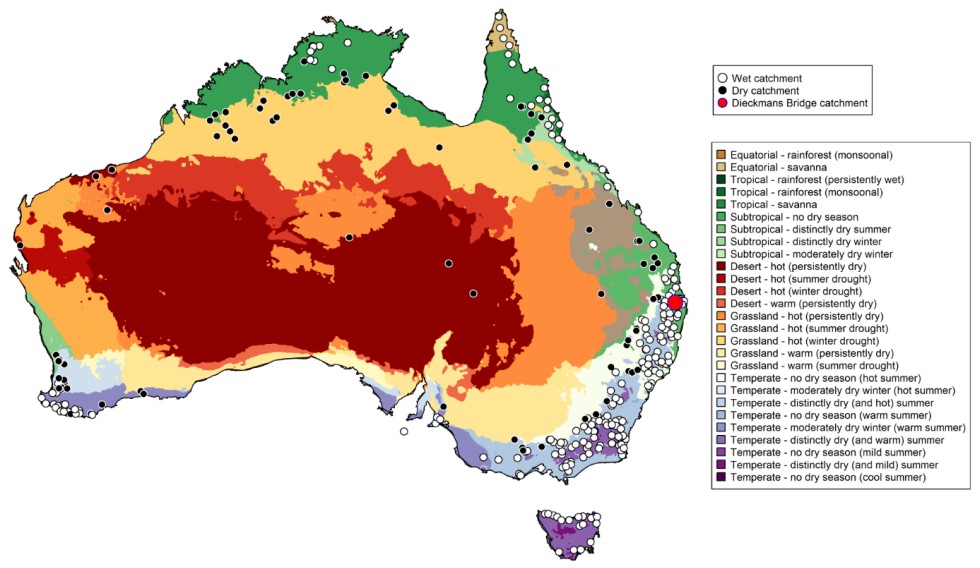


Figure 2: Locations of the 300 catchments used in this study. The catchments are classified as dry or wet based on the aridity index. The Koppen climate classification for Australia are shown. The Dieckmans Bridge catchment (site id: 145010A), used as a representative site in Figure 8, is indicated by the red circle.
















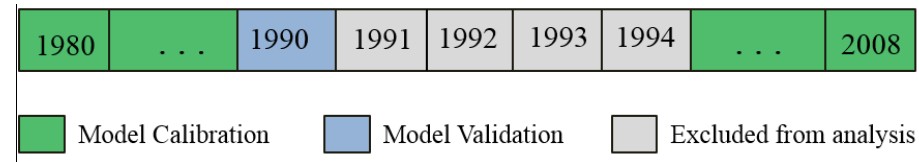


Figure 3: Schematic of the cross-validation framework used for forecast verification as an example for
model validation year 1990 (after Tuteja et al., 2016).















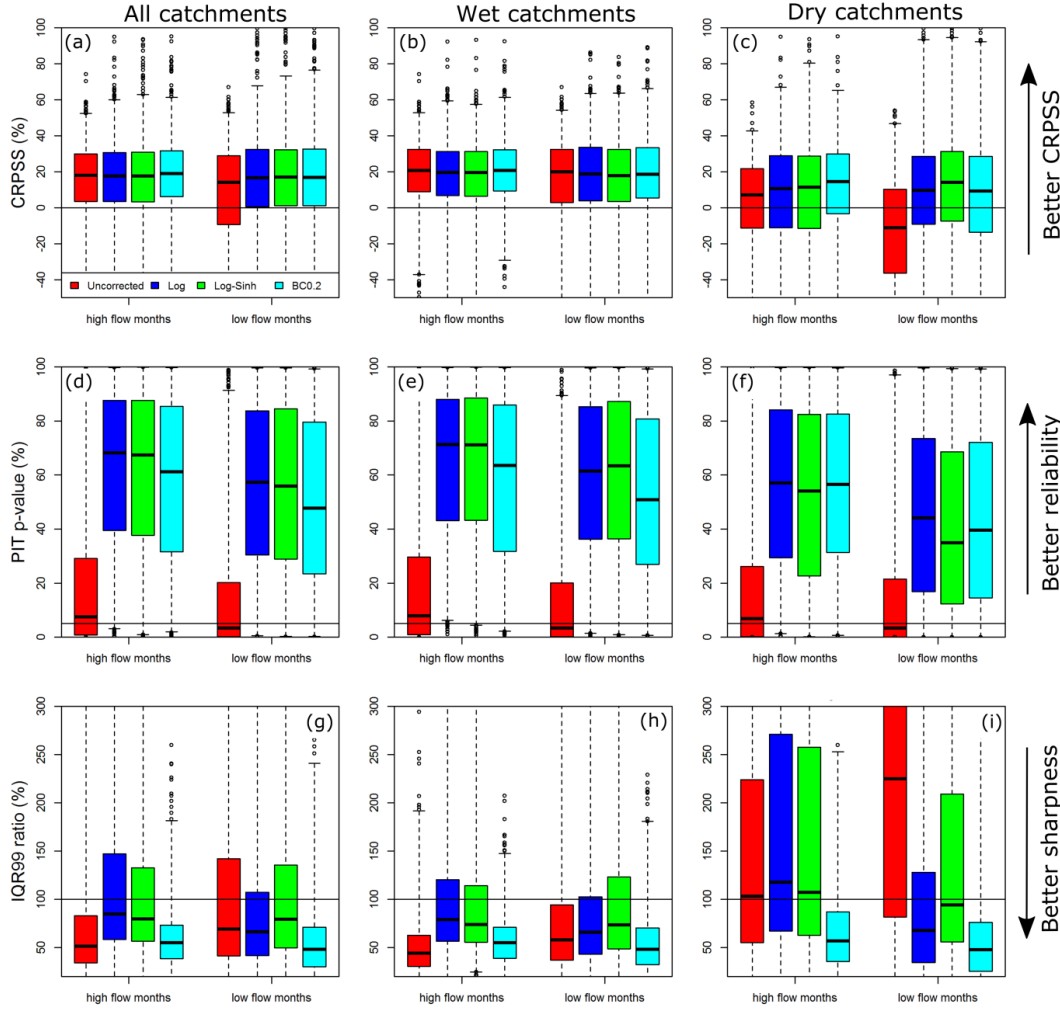


Figure 4: Performance of monthly forecasts in terms of CRPSS, reliability (PIT p-value) and sharpness
(IQR99 ratio).




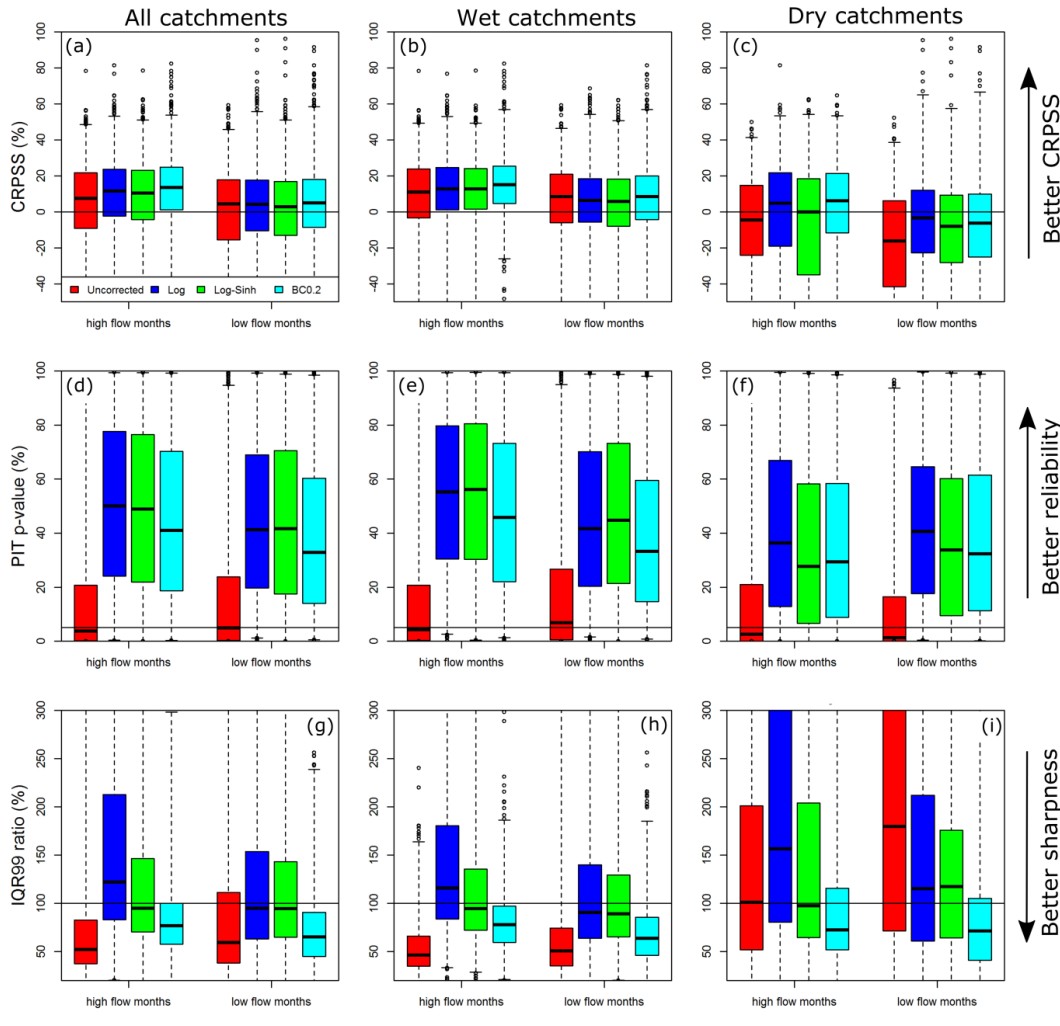


Figure 5: Performance of seasonal forecasts in terms of CRPSS, reliability (PIT p-value) and sharpness
(IQR99 ratio).



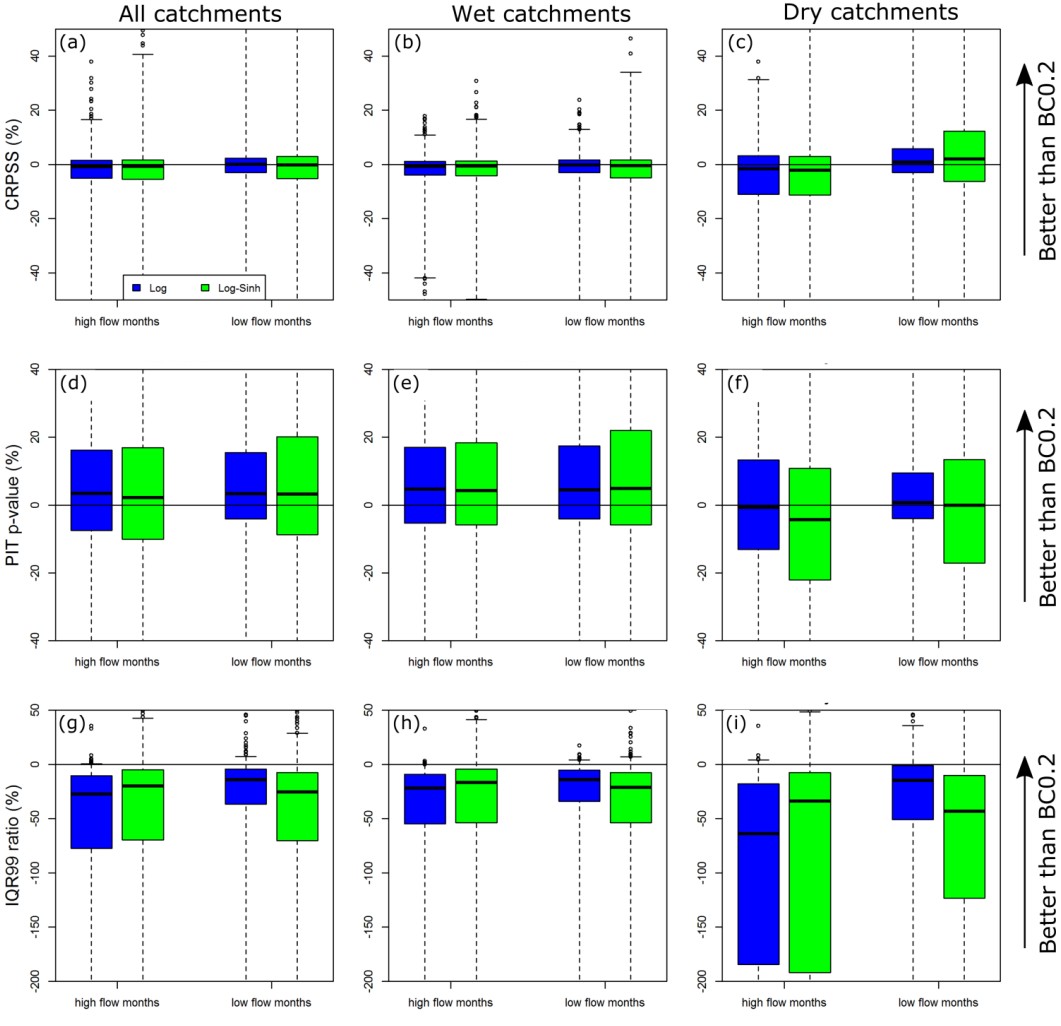


Figure 6: Distributions of differences in the monthly forecast performance metrics of the Log and Log-
Sinh schemes compared to the BC0.2 scheme.







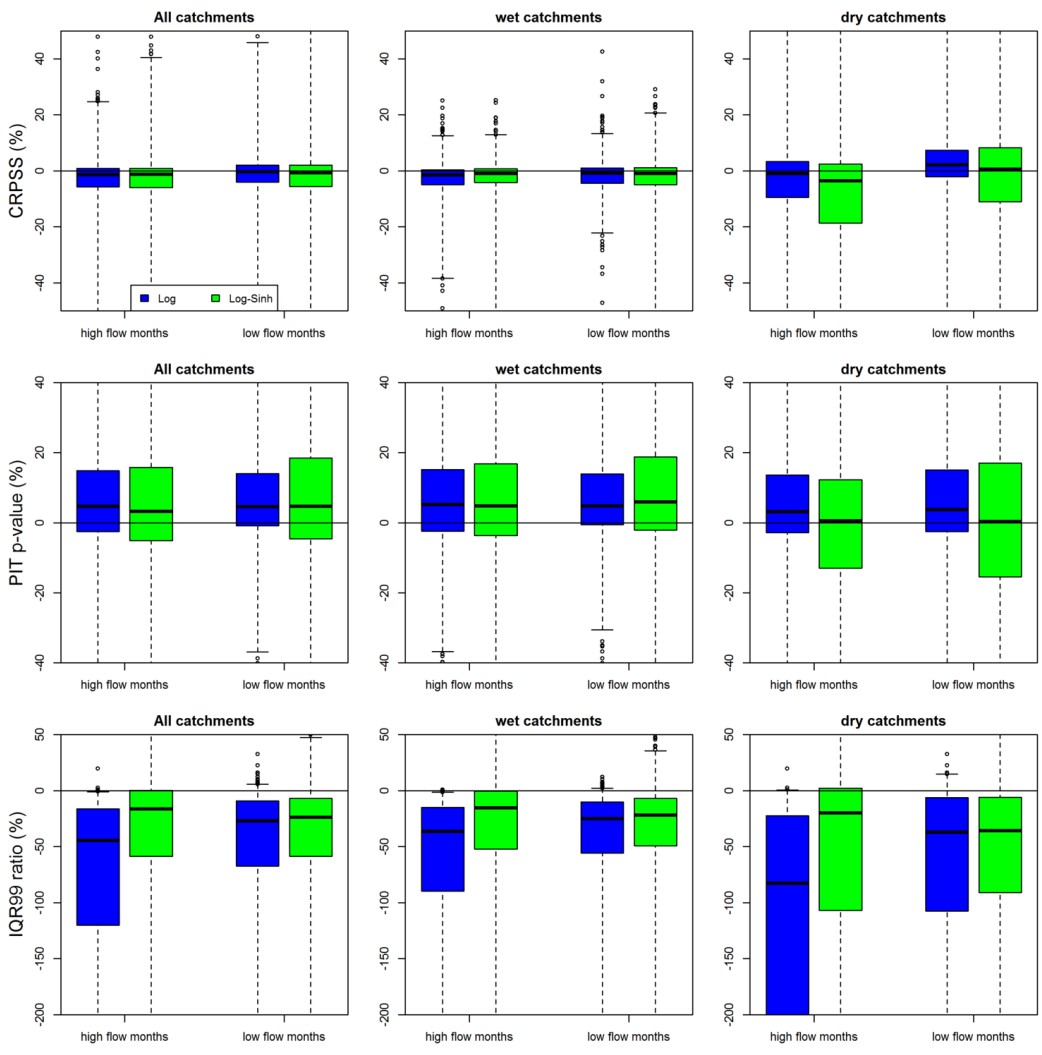


Figure 7: Distributions of differences in the seasonal forecast performance metrics of the Log and Log-Sinh schemes compared to the BC0.2 scheme.








Figure 8: Seasonal streamflow forecast time series (blue line) and observations (red dots) at Dieckmans
Bridge catchment (site id: 145010A). The shaded area shows the 99% prediction limits.





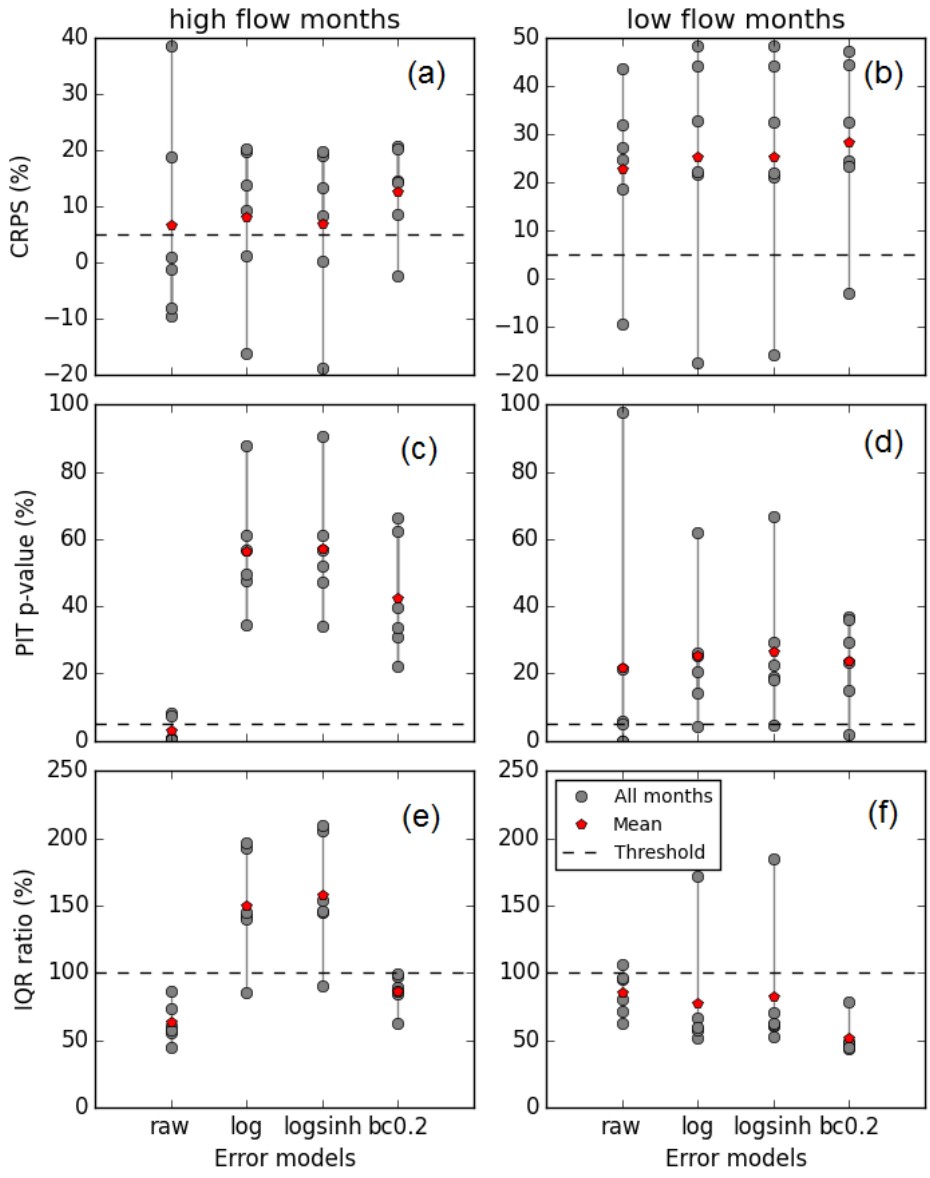


Figure 9: Seasonal streamflow forecast skill-score at the Dieckmans Bridge catchment corresponding to
the time series shown in Figure 8 for six high flow months and six low flow months. Note that skill-
score values of 5%, 5% and 100% are indicated for CRPSS, p-value and IQR ratio respectively, using
dashed lines.






Figure 10: Summary skill of monthly forecasts obtained using the Log, Log-Sinh and BC0.2 schemes across 300 Australian catchments. The performance of uncorrected forecasts is also shown. The summary skill is defined as the number of months where the forecasts are reliable and sharper than climatology. The inset histogram shows the percentage of catchments in each performance category and also serves as the color legend.






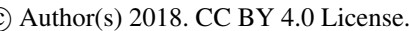

Figure 11: Summary skill of seasonal forecasts obtained using the Log, Log-Sinh and BC0.2 schemes
across 300 Australian catchments. See Figure 10 for details.




