# Peer review of "Evaluating post-processing approaches for monthly and seasonal streamflow forecasts"

_Hydrology and Earth System Sciences, 2018_

## Referee Comment (RC1) · Anonymous Referee #1 · 25 May 2018

This review is for Manuscript ID: hess-2018-214, entitled Evaluating Residual Error Approaches for Post-processing Monthly and Seasonal Streamflow Forecasts, authored by Fitsum Woldemeskel and coauthors. With this manuscript the authors' aim is to evaluate different residual error models, including logarithmic (Log), Log-Sinh, and Box-Cox transformation schemes, for postprocessing monthly and seasonal streamflow forecasts. Overall, the postprocessed streamflow forecasts demonstrate skillful, reliable and sharper forecasts compared to the uncorrected forecasts. Furthermore, postprocessor employing the Box-Cox transformation scheme demonstrate the sharpest forecasts, without sacrificing skill and reliability. This manuscript is generally clear, however, it reads like a book chapter rather than a journal article. I believe the results

and conclusions are of interest to the HESS community, as well as to the operational forecasters. Thus, this manuscript is worthy of publication if the issues below are addressed.

Major Comments

1) The introduction needs better organization. Consider removing the unnecessary details about the statistical modelling system and hybrid system (P4-5, L86-95), which are irrelevant in the context of dynamic modeling. The literature review can be focused on the usefulness of POAMA-2 in advancing seasonal hydrological forecasting.

2) Make a separate subsection for the study area, dataset and hydrological model.

- Study area: Provide general information on the hydroclimatic conditions, types of events across different seasons, basin size range, and reason for selecting the particular catchments.

- Dataset: Provide detail information on rainfall forecast dataset from POAMA-2, including forecast lead time, total number of ensemble members, and forecast initialization time and frequency. POAMA-2 information (P7, L189-194) should be integrated into the "Section 3.1 Data".

- Hydrological model: I am concerned about the details of the rainfall-runoff model GR4J used for the study. It is necessary that you explain better the following aspects of the model: lumped conceptual model or physically based model, spatial resolution of the model, and the selected routing method. How often is the model initialized to make the forecast runs?

3) If the model is calibrated, then consider adding a subsection to discuss the simulation performance. You need to mention the calibrated parameters, model warm-up period, calibration period and validation period. The simulation performance can be discussed using correlation coefficient, percent bias and Nash-Sutcliffe efficiency between the observed and simulated streamflow.
[Figure]

4) In order to support the operational forecasting system, the conclusions drawn here should be valid in the context of extreme events. Does the conclusions apply to flood events? For this, verification metrics can be computed by considering the flow amounts greater than that implied by a non-exceedance probability, in the sampled climatological probability distribution, of 0.95.

5) Considering an operational forecasting situation, how feasible is it to run 166 ensemble members using 40 GR4J parameters, and produce 6640 daily streamflow forecasts?

6) In the context of seasonal forecasting, different studies have demonstrated the combined ability of preprocessing meteorological forcing and postprocessing streamflow forecast to produce better streamflow forecasts. However, the study here only implements postprocessing. Was the meteorological forcing preprocessed? If not the case, it could be a topic of discussion, as a recommendation for future work to investigate the performance of residual error models in the context of preprocessing and postprocessing.

Minor Comments

1) Figure 8: Mention the units in the Y-axis for streamflow.

2) Figure 8: Is there any reason for selecting Dieckmans Bridge catchment as a representative site for the analysis. Why is the time series plotted only for the period of 2003-2007? Is this a random selection?

3) Figure 9a: Replace "CRPS" with "CRPSS" in the Y-axis.

4) P8 L200-204: Integrate this paragraph into the introduction.

5) P9 L233: Provide a reference to the statement: "the parameters are estimated based on the methods of moments."

6) P13 L365: Define the variable "y" in Equation 11.

7) P13 L367: How do you define the Heaviside step function?

8) P16 L444: Fix the typo for "Figure 45i".

9) P18 L495: Replace "unprocessed" with" uncorrected".

10) P18 L501: Define the acronyms: "NSW", "QLD" and "NT" when used for the first time.

11) It may be good idea to provide a standard name for the streamflow postprocessing technique implemented in the study, is it a new technique? If not, then provide a suitable reference to the postprocessing technique.

Please also note the supplement to this comment:
https://www.hydrol-earth-syst-sci-discuss.net/hess-2018-214/hess-2018-214-RC1-supplement.pdf

---

## Referee Comment (RC2) · Anonymous Referee #2 · 31 May 2018

This paper presents a comparison of three variants of a post-processing approach for long-range (here monthly to seasonal) streamflow forecasts in Australia. The paper is well-written and easy to read. The research is interesting for several reasons. First, the topic of long-range forecasting and especially how the skill of such forecasts can be improved is currently raising a lot of attention from hydrologists. There are many practical applications for which seasonal forecasts are required for decision-making. Second, Australia is a vast country which includes a broad range of hydro-climatic conditions, and the authors efficiently gathered a data base of 300 catchments, to ensure that their findings are as generalizable as possible. The authors explicitly address the specific case of dry catchments and low-flow periods, which is of great practical interest in sev-

eral areas on the planet. Indirectly, the research has implications for socio-economic issues, as the management of water-scarce catchments could benefit from better seasonal forecasts. The paper also fits well within the scope of HESS.

This paper is definitely suitable for publication in HESS. However, I do have a few specific comments and suggestions for the authors to improve it prior to publication. In my opinion, all comments are minor.

Specific comments/suggestions/questions:

1. Data assimilation/link with short-term forecasts

From the paper, I am not entirely sure how GR4J's state variables are managed (see also my next comment). From what I understand, there is no data assimilation at all. Perhaps this can be justified in the context of long-range forecasts as the effect of data assimilation would fade out quickly (probably before the one month horizon).

I was also wondering if there is a link (operationaly at BoM) between short- and long-range forecasts. Surely the hydrological model is the same, but what about the meteorological forecasts? Are the short- and long-ranges connected in some way? Surely, in operations, there must be a certain form of data assimilation for short lead times.

Considering the above interrogations, I would very much appreciate short comment regarding data assimilation in the paper.

2. Simulation and Forecast steps vs model calibration and warm-up

Section 2.2 and 2.3: I am slightly puzzled by that division into "simulation step", which includes model calibration, and "forecast step". Reading the description of the "simulation step", one could thing that you re-calibrate the model several times, once before each forecasting step. Is that so? If so, why? You want the model parameters to be dynamic?

I would tend to think that what the steps would rather be (1) calibrate the model (40

times using the MCMC-based method you mention) once and for all, (2) simulate streamflow over the entire period and save the state variables (I assume no data assimilation) and (3) launch GR4J in "forecast mode", by fetching the appropriate state variables for a specific date and feeding the model with meteorological forecasts.

I would very much appreciate if you could clarify those issues in the paper. In particular, think that calibration should be separated from simulation.

3. Discussing the choice of model for dry catchments

Section 5, lines 535-536, you mention that "This finding can be attributed to the challenge of capturing key physical processes in modeling dry and ephemeral catchments (. . .)". In my opinion, this sentence leads to questioning whether of not GR4J is an appropriate model for very dry catchments. I know this model very well and I can appreciate its many qualities. GR4J works well for a very wide variety of hydro-climatic conditions. In addition, I do understand the practicality of having only one (very simple) model for all catchments on the entire country. However, there is no soil per se in GR4J. It is a very simple conceptual model which cannot, for instance, model soil sealing phenomena for dry catchments. I don't see how this model could ever capture the physical processes, as mentioned in your sentence.

In my opinion, this issue (the choice of a very simple conceptual model) should be briefly discussed following lines 535-536.

4. Citing papers from HESS Discussion

In my opinion, citing papers from HESS Discussion should be discouraged. After all, there is no real filtering of the papers before they can be published in discussion. The revision process takes place around the Discussion paper. To me, a paper that never makes it to HESS (after the Discussion) should be considered as rejected, even though it remains publicly accessible on the web. You wouldn't cite a paper that was rejected from other "more traditional" journals for which the revision is not as public as for HESS.

Of course you could argue that if a paper in Discussion receives excellent comments but never makes it to HESS, it could be a case where the authors purposefully decided not to spend time editing it according to the reviewer's comments and re-submitting it. In my opinion, this practice, if it exists, should not be encouraged. Again, it wouldn't be possible with the majority of other journals.

Therefore, I would very strongly recommend that you remove all references to HESS Discussion. Set (2006) should therefore not be cited.

The citation for Mendoza et al (2017) should be updated as it is now published. Same for Turner et al (2017). The titles have also changed in the published version.

5. Forecasts' value

Section 5.3 lines 584-587, you briefly touch on the issue of forecasts value. I personally don't think measures of skill could ever be linked to the socio-economic value of forecasts. Most studies focussing on forecast values in the current literature largely over-simplify the problem. For the issue of forecasts value to be tackles in a more realistic way, researchers from humanities and social sciences would inevitably have to be involved. Forecasts value involves complex issues related to human psychology, economic theory, communication, social studies, etc. See for instance Morss et al. (2010), Matte et al. (2017), Toon et al. (2017) and Solin et al (2018)

In my opinion, forecasts skill is a pre-requisite for forecast value but in no way a guaranty. I don't see how metrics related strictly to the skill of a forecast (as in comparing the forecast to observation) could be a predictor of forecasts value on their own.

Morss et al (2017) Examining the use of weather forecasts in decision scenarios: results from a US survey with implications for uncertainty communication, METEOROLOGICAL APPLICATIONS, 17(2), 149-162

Matte et al (2017) Moving beyond the cost-loss ratio: economic assessment of streamflow forecasts for a risk-averse decision maker, HYDROLOGY AND EARTH SYSTEM

SCIENCES, 21 (6), 2967-2986.

Toon et al (2017) Integrating Household Risk Mitigation Behavior in Flood Risk Analysis: An Agent-Based Model Approach, RISK ANALYSIS, 37 (10), 1977-1992

Solin et al (2018) Vulnerability assessment of households and its possible reflection in flood risk management: The case of the upper Myjava basin, Slovakia, INTERNATIONAL JOURNAL OF DISASTER RISK REDUCTION, 28, 640-652

6. Typos/spelling/format/figures

- Page 10 line 255: I think the word " trial" should be replaced by "tried".

- Page 13 equation 11: The CRPS is usually computed by averaging the values over a large sample of forecasts-observation groups. Therefore, I think it is important that equation (11) be modified to be more explicit about this averaging.

- Page 14 line 388: "lead to misleading" is a bit strange to read. I would advise rephrasing

- Page 15 lines 413-414: there seem to be an awkward space between those two lines. Please verify.

- Page 16 lines 443-444: Is "from in excess of 150%" the correct phrasing? Also, there is a typo in the parenthesis "(Figure 45i)".

- Page 18 line 493: remove comma after "scheme"

- Page 37, figure 8: please include the units for streamflow (y axis) on this figure. In addition, I am not entirely sure I understand the time step (x axis). Counting the points, I understand that the time step is one month, which would be coherent with the text, but not explicitly specified for this figure. In my opinion the x axis label could also be clearer.

- Page 38 figure 9: An "S" is missing for the y axis label of the top row. It should be

CRPSS and not CRPS.

---

## Short Comment (SC1) · 16 Jun 2018

Response to Referee #1 General comment: This review is for Manuscript ID: hess-2018-214, entitled Evaluating Residual Error Approaches for Post-processing Monthly and Seasonal Streamflow Forecasts, authored by Fitsum Woldemeskel and coauthors. With this manuscript the authors' aim is to evaluate different residual error models, including logarithmic (Log), Log-Sinh, and Box-Cox transformation schemes, for post-processing monthly and seasonal streamflow forecasts. Overall, the postprocessed streamflow forecasts demonstrate skillful, reliable and sharper forecasts compared to the uncorrected forecasts. Furthermore, postprocessor employing the Box-Cox trans-

formation scheme demonstrate the sharpest forecasts, without sacrificing skill and reliability. This manuscript is generally clear, however, it reads like a book chapter rather than a journal article. I believe the results and conclusions are of interest to the HESS community, as well as to the operational forecasters. Thus, this manuscript is worthy of publication if the issues below are addressed.

Author response: We thank the reviewer for the positive assessment of our manuscript as well as for their constructive comments and useful suggestions to improve the manuscript further. We are pleased that the reviewer found our manuscript suitable for the HESS research community and the community of operational forecasters. We provide specific responses to review comments as follows.

Major Comments

Referee comment 1: The introduction needs better organization. Consider removing the unnecessary details about the statistical modelling system and hybrid system (P4-5, L86-95), which are irrelevant in the context of dynamic modeling. The literature review can be focused on the usefulness of POAMA-2 in advancing seasonal hydrological forecasting.

Author response: We agree that statistical and hybrid systems are not directly relevant in the context of dynamic modelling. In the revised version, we will ensure the description of statistical and hybrid approaches focus on the essentials and avoid excessive details. We will also elaborate on the usefulness of rainfall forecasts, including POAMA-2, for streamflow forecasting.

Referee comment 2: Make a separate subsection for the study area, dataset and hydrological model.

a) Study area: Provide general information on the hydroclimatic conditions, types of events across different seasons, basin size range, and reason for selecting the particular catchments.

[Figure]

Author response: Thank you for this suggestion. We will add separate subsection for study area with additional details about the catchments studied. As we are evaluating 300 catchments, it will be difficult to provide detailed site specific information, however, we intend to provide summarised information highlighting those points suggested by the reviewer.

b) Dataset: Provide detail information on rainfall forecast dataset from POAMA-2, including forecast lead time, total number of ensemble members, and forecast initialization time and frequency. POAMA-2 information (P7, L189-194) should be integrated into the "Section 3.1 Data".

Author response: We will add additional information about rainfall forecast using POAMA-2 as well as integrate the POAMA-2 information (p7, L189-194) into Section 3.1.

c) Hydrological model: I am concerned about the details of the rainfall-runoff model GR4J used for the study. It is necessary that you explain better the following aspects of the model: lumped conceptual model or physically based model, spatial resolution of the model, and the selected routing method. How often is the model initialized to make the forecast runs?

Author response: Thanks for raising this issue. In the revised manuscript we will include additional details about the GR4J model.

Referee comment 3: If the model is calibrated, then consider adding a subsection to discuss the simulation performance. You need to mention the calibrated parameters, model warm-up period, calibration period and validation period. The simulation performance can be discussed using correlation coefficient, percent bias and Nash-Sutcliffe efficiency between the observed and simulated streamflow.

Author response: Yes we have calibrated the GR4J model parameters. We use 5 years model warm-up during 1975-1979 as well as calibration and validation during

1980-2008 in a moving 5 years leave-out cross-validation scheme. We will make these points clearer in the revised manuscript.

In the paper, we have focused on forecast performance as this is the operational goal of the Bureau of Meteorology. In addition, we are using the same calibrated hydrological model for the three error modelling schemes, thus the results and conclusions regarding the error model schemes remain the same regardless of the simulation performance. Considering these and for conciseness, we will limit the analysis and results on forecast performance.

Referee comment 4: In order to support the operational forecasting system, the conclusions drawn here should be valid in the context of extreme events. Does the conclusions apply to flood events? For this, verification metrics can be computed by considering the flow amounts greater than that implied by a non-exceedance probability, in the sampled climatological probability distribution, of 0.95.

Author response: While seasonal streamflow forecasts have limited application for flood prediction purposes, the question is relevant for predicting drought events, where the seasonal forecasts have significant value. In this study we evaluated forecast performance separately for high and low flow months, which provides an indication of predictive ability for below-average flows (i. e., drought events). In addition, the results and conclusion regarding the best performing error model scheme and its performance apply for the extreme events.

Having said that, robust evaluation of forecasts at extreme events (e.g. drought events only) is challenging as these events are rare. Limited sample size makes it difficult to make conclusive statements. For instance, with a full record of 30 years used for calibration, if we want to test against <5% or >95% of historical data, we might have only roughly 1.5 samples to be tested for each month/season, which will add high uncertainty to the verification results and make it difficult to draw definitive conclusions. To handle this uncertainty requires the development of new forecast verification techniques potentially adapting some of the approaches for groups of catchments used by Hodgkins et al. (2017). As the development of new forecast verification techniques is outside the scope of this current study, we will include a paragraph in the discussion to acknowledge this issue and recommend it for future investigations.

Referee comment 5: Considering an operational forecasting situation, how feasible is it to run 166 ensemble members using 40 GR4J parameters, and produce 6640 daily streamflow forecasts?

Author response: Yes, it is feasible to run 166 ensemble members with 40 GR4J parameters, and the Bureau of Meteorology has been running such a system operationally for a few years now. Producing 6640 forecasts this way is important to maintain reliability of forecasts. The largest computational expense results from calibrating hydrological models and cross-validation exercise rather than updating streamflow forecasts once every month using 166 ensembles members. However, the calibration and cross-validation exercise is typically done using a single observed rainfall time-series. We also use high performance computing (HPC) facilities available at the Bureau of Meteorology and the National Computing Infrastructure (NCI) for calibrating hydrological models, which significantly reduces overall computation time. We will mention this in the revised manuscript.

Referee comment 6: In the context of seasonal forecasting, different studies have demonstrated the combined ability of preprocessing meteorological forcing and post-processing streamflow forecast to produce better streamflow forecasts. However, the study here only implements postprocessing. Was the meteorological forcing preprocessed? If not the case, it could be a topic of discussion, as a recommendation for future work to investigate the performance of residual error models in the context of preprocessing and postprocessing.

Author response: We use the analogue approach to downscale gridded POAMA-2 rainfall forecast to catchment scale forecast, which can be considered as some form of

pre-processing. We will highlight this point in the revised manuscript.

Minor Comments

Referee comment 7: Figure 8: Mention the units in the Y-axis for streamflow.

Author response: Thank you for this suggestion. We will include units in the Y-axis in Figure 8.

Referee comment 8: Figure 8: Is there any reason for selecting Dieckmans Bridge catchment as a representative site for the analysis. Why is the time series plotted only for the period of 2003-2007? Is this a random selection?

Author response: Dickmans Bridge catchment is selected as it is reflective of the results and conclusions across all catchments. That is, applying BC0.2 at this catchment resulted in better sharpness compared to applying Log and LogSinh while maintaining comparable CRPSS and reliability for high and low flow months. This is shown in Figure 9. The period 2003-2007 in Figure 8 is chosen as this period shows the difference in the forecast interval between the raw and three error models more clearly. We will highlight this point in the revised manuscript.

Referee comment 9: Figure 9a: Replace "CRPS" with "CRPSS" in the Y-axis. Referee comment 10: P8 L200-204: Integrate this paragraph into the introduction. Referee comment 11: P9 L233: Provide a reference to the statement: "the parameters are estimated based on the methods of moments." Referee comment 12: P13 L365: Define the variable "y" in Equation 11. Referee comment 13: P13 L367: How do you define the Heaviside step function? Referee comment 14: P16 L444: Fix the typo for "Figure 45i". Referee comment 15: P18 L495: Replace "unprocessed" with" uncorrected". Referee comment 16: P18 L501: Define the acronyms: "NSW", "QLD" and "NT" when used for the first time.

Author response: We thank the reviewer for pointing out the above editorial corrections (comments 9-16). We will incorporate these corrections in the revised manuscript.

[Figure]

Referee comment 17: It may be good idea to provide a standard name for the stream-flow postprocessing technique implemented in the study, is it a new technique? If not, then provide a suitable reference to the postprocessing technique.

Author response: We thank the reviewer for this suggestion. The residual error model approach used in this study is not new (e.g. the Box-Cox / power transformation has been introduced by Box and Cox, 1964; see McInerney et al., 2017 for detailed analysis), however, the application of it for post-processing monthly and seasonal streamflow forecasting in national forecasting system is new. We will clarify this in the revised manuscript.

References

Box, G. E. P. and Cox, D. R.: An analysis of transformations, Journal of the Royal Statistical Society, Series B. 26 (2): 211–252. JSTOR 2984418. MR 0192611.

Hodgkins, G. A., P. H. Whitfield, D. H. Burn, J. Hannaford, B. Renard, K. Stahl, A. K. Fleig, H. Madsen, L. Mediero, J. Korhonen, C. Murphy, and D. Wilson.: Climate-driven variability in the occurrence of major floods across North America and Europe, J Hydrol, 552, 704-717, 10.1016/j.jhydrol.2017.07.027.

McInerney, D., Thyer, M., Kavetski, D., Lerat, J. and Kuczera, G.: Improving probabilistic prediction of daily streamflow by identifying Pareto optimal approaches for modeling heteroscedastic residual errors, Water Resour. Res., 53(3), 2199–2239, doi:10.1002/2016WR019168, 2017.

---

## Short Comment (SC2) · 16 Jun 2018

Response to Referee #2 General comment: This paper presents a comparison of three variants of a post-processing approach for long-range (here monthly to seasonal) streamflow forecasts in Australia. The paper is well-written and easy to read. The research is interesting for several reasons. First, the topic of long-range forecasting and especially how the skill of such forecasts can be improved is currently raising a lot of attention from hydrologists. There are many practical applications for which seasonal forecasts are required for decision-making. Second, Australia is a vast country which includes a broad range of hydro-climatic conditions, and the authors efficiently gath-

ered a data base of 300 catchments, to ensure that their findings are as generalizable as possible. The authors explicitly address the specific case of dry catchments and low-flow periods, which is of great practical interest in several areas on the planet. Indirectly, the research has implications for socio-economic issues, as the management of water-scarce catchments could benefit from better seasonal forecasts. The paper also fits well within the scope of HESS.

This paper is definitely suitable for publication in HESS. However, I do have a few specific comments and suggestions for the authors to improve it prior to publication. In my opinion, all comments are minor.

Author response: We thank the reviewer for this very positive assessment of our manuscript and for recognising its practical importance. We are pleased that reviewer found our manuscript suitable for publication in HESS. We provide response to the review comments as follows.

Specific comments/suggestions/questions:

Referee comment 1: Data assimilation/link with short-term forecasts

From the paper, I am not entirely sure how GR4J's state variables are managed (see also my next comment). From what I understand, there is no data assimilation at all. Perhaps this can be justified in the context of long-range forecasts as the effect of data assimilation would fade out quickly (probably before the one month horizon).

I was also wondering if there is a link (operationaly at BoM) between short- and longrange forecasts. Surely the hydrological model is the same, but what about the meteorological forecasts? Are the short- and long-ranges connected in some way? Surely, in operations, there must be a certain form of data assimilation for short lead times.

Considering the above interrogations, I would very much appreciate short comment regarding data assimilation in the paper.

Author response: The reviewer is correct that we have not used data assimilation to up-

date GR4J state variables. However, data assimilation of ocean observations has been incorporated in the climate model (POAMA2.0) from which the rainfall forecasts have been obtained. We agree with the reviewer that the benefit of data assimilation for seasonal forecasts is limited. However, Gibbs et al. (2018) showed that monthly streamflow forecasting could benefit from state updating when the issue of non-stationarity is also handled. This is something to be investigated further in the future. We will provide a short comment about this issue in the manuscript.

Regarding the link between short- and long range forecasts provided by the Bureau of Meteorology, the two have independent systems due to the different needs of the forecasting service stakeholders. Short-range forecasts require a daily update with a focus on timely delivery of forecasts to anticipate quickly evolving events. Long-range forecast are more connected to longer-term decision making, which requires monthly update and statistically reliable forecasts. In addition, the GCM inputs are sourced from different models: short-term streamflow forecasts use weather forecasting model with limited modelling of ocean dynamic, whereas long range streamflow forecasts use climate outlook model with strong coupling between ocean and atmospheric models. As of now, updating state variables through data assimilation is also not yet implemented in short-range forecasts, but there are plans to incorporate this in the future.

Overall, the streamflow forecast relies on data assimilation included in the climate model and of robust hydrologic modelling technique highlighted in the paper. It is also worth to mention that the Bureau of Meteorology prioritised investments in developing hydrologic modelling within a robust uncertainty framework, followed by streamflow and rainfall post-processing. In our view, the incremental benefits from data assimilation is likely to be less than these components.

Referee comment 2: Simulation and Forecast steps vs model calibration and warm-up

Section 2.2 and 2.3: I am slightly puzzled by that division into "simulation step", which includes model calibration, and "forecast step". Reading the description of the "simulation step", one could thing that you re-calibrate the model several times, once before each forecasting step. Is that so? If so, why? You want the model parameters to be dynamic?

I would tend to think that what the steps would rather be (1) calibrate the model (40 times using the MCMC-based method you mention) once and for all, (2) simulate streamflow over the entire period and save the state variables (I assume no data assimilation) and (3) launch GR4J in "forecast mode", by fetching the appropriate state variables for a specific date and feeding the model with meteorological forecasts.

I would very much appreciate if you could clarify those issues in the paper. In particular, think that calibration should be separated from simulation.

Author response: The three steps the reviewer described are correct. However, there is an additional process in step 1, i.e., we estimate parameters in a moving 5 years leave-out cross-validation approach. This is done in order to validate forecasts with an observed data independent from the ones used for calibration. When we use moving 5 years leave-out cross-validation scheme, the parameters would be slightly different for each year.

We use data from 1980-2008 for cross-validation with a model warm-up period of 5 years (i.e. 1975-1979).

We will further clarify these points in the paper as well as make the distinction between calibration and simulation clearer as suggested by the reviewer.

Referee comment 3: Discussing the choice of model for dry catchments

Section 5, lines 535-536, you mention that "This finding can be attributed to the challenge of capturing key physical processes in modeling dry and ephemeral catchments (: : :)". In my opinion, this sentence leads to questioning whether of not GR4J is an appropriate model for very dry catchments. I know this model very well and I can appreciate its many qualities. GR4J works well for a very wide variety of hydro-climatic

conditions. In addition, I do understand the practicality of having only one (very simple) model for all catchments on the entire country. However, there is no soil per se in GR4J. It is a very simple conceptual model which cannot, for instance, model soil sealing phenomena for dry catchments. I don't see how this model could ever capture the physical processes, as mentioned in your sentence.

In my opinion, this issue (the choice of a very simple conceptual model) should be briefly discussed following lines 535-536.

Author response: We thank the reviewer for pointing out this issue. The reviewer is correct that the model structure of GR4J, in particular its simplifying assumptions, might be responsible for the relatively lower forecast skill in dry catchments as compared to wet. Having said that from our experience, uncertainty of rainfall forecast dwarfs the hydrologic uncertainty. Our intent with respect to hydrological modelling is to use a model that can perform as best as possible in different hydro-climate conditions without necessarily being complicated and non-parsimonious. Whilst using a single simple conceptual model is attractive for a practical operational system, there may be gains in exploring alternative model structures for difficult catchments (e.g. Clark et al., 2008; Fenicia et al., 2011). We intend to explore alternative model structures for difficult ephemeral catchments. We will elaborate on these issues on lines 535-536 as well as highlight some of these points further in the revised manuscript.

It is also worth mentioning that forecasting in dry catchments will remain an issue regardless of the hydrological model used due to the limited amount of information contained in streamflow records (high number of zero flow values) and high frequency of convective storms.

Referee comment 4: Citing papers from HESS Discussion

In my opinion, citing papers from HESS Discussion should be discouraged. After all, there is no real filtering of the papers before they can be published in discussion. The revision process takes place around the Discussion paper. To me, a paper that never

makes it to HESS (after the Discussion) should be considered as rejected, even though it remains publicly accessible on the web. You wouldn't cite a paper that was rejected from other "more traditional" journals for which the revision is not as public as for HESS. Of course you could argue that if a paper in Discussion receives excellent comments but never makes it to HESS, it could be a case where the authors purposefully decided not to spend time editing it according to the reviewer's comments and re-submitting it. In my opinion, this practice, if it exists, should not be encouraged. Again, it wouldn't be possible with the majority of other journals.

Therefore, I would very strongly recommend that you remove all references to HESS Discussion. Set (2006) should therefore not be cited.

The citation for Mendoza et al (2017) should be updated as it is now published. Same for Turner et al (2017). The titles have also changed in the published version.

Author response: We agree with the suggestion not to cite HESS Discussion papers. Therefore, we will remove or modify the above references as appropriate in the revised manuscript. We will also update the references as suggested.

Referee comment 5: Forecasts' value

Section 5.3 lines 584-587, you briefly touch on the issue of forecasts value. I personally don't think measures of skill could ever be linked to the socio-economic value of forecasts. Most studies focussing on forecast values in the current literature largely over-simplify the problem. For the issue of forecasts value to be tackles in a more realistic way, researchers from humanities and social sciences would inevitably have to be involved. Forecasts value involves complex issues related to human psychology, economic theory, communication, social studies, etc. See for instance Morss et al. (2010), Matte et al. (2017), Toon et al. (2017) and Solin et al (2018).

In my opinion, forecasts skill is a pre-requisite for forecast value but in no way a guaranty. I don't see how metrics related strictly to the skill of a forecast (as in comparing

the forecast to observation) could be a predictor of forecasts value on their own.

Morss et al (2017) Examining the use of weather forecasts in decision scenarios: results from a US survey with implications for uncertainty communication, METEOROLOGICAL APPLICATIONS, 17(2), 149-162

Matte et al (2017) Moving beyond the cost-loss ratio: economic assessment of streamflowforecasts for a risk-averse decision maker, HYDROLOGY AND EARTH SYSTEM SCIENCES, 21 (6), 2967-2986.

Toon et al (2017) Integrating Household Risk Mitigation Behavior in Flood Risk Analysis: An Agent-Based Model Approach, RISK ANALYSIS, 37 (10), 1977-1992

Solin et al (2018) Vulnerability assessment of households and its possible reflection in flood risk management: The case of the upper Myjava basin, Slovakia, INTERNATIONAL JOURNAL OF DISASTER RISK REDUCTION, 28, 640-652.

Author response: We thank the reviewer for these insights on the value of forecasts as well as for the suggestion of relevant literatures. We agree with the reviewer that forecast skill is a pre-requisite but not a guarantee of its value. From an operational point of view, having a way to link skill and value would be highly valuable. The Bureau actively woks with its stakeholders to provide evidence about this point by developing forecast application case studies (http://www.bom.gov.au/water/ssf/case_studies.shtml). A socio-economic study conducted by London Economics has also highlighted the value of seasonal forecasts, among other Bureau services (Duke et al. 2016). However, a link between skill and value is a very complex issue as mentioned by the reviewer. We will modify the text in line 584-587 to briefly elaborate on these aspects.

Referee comment 6: Typos/spelling/format/figures

- Page 10 line 255: I think the word " trial" should be replaced by "tried". - Page 13 equation 11: The CRPS is usually computed by averaging the values over a large sample of forecasts-observation groups. Therefore, I think it is important that equation

(11) be modified to be more explicit about this averaging. - Page 14 line 388: "lead to misleading" is a bit strange to read. I would advise rephrasing - Page 15 lines 413-414: there seem to be an awkward space between those two lines. Please verify. - Page 16 lines 443-444: Is "from in excess of 150%" the correct phrasing? Also, there is a typo in the parenthesis "(Figure 45i)". - Page 18 line 493: remove comma after "scheme" - Page 37, figure 8: please include the units for streamflow (y axis) on this figure. In addition, I am not entirely sure I understand the time step (x axis). Counting the points, I understand that the time step is one month, which would be coherent with the text, but not explicitly specified for this figure. In my opinion the x axis label could also be clearer. - Page 38 figure 9: An "S" is missing for the y axis label of the top row. It should be CRPSS and not CRPS.

Author response: We thank the reviewer for pointing out the above editorial corrections. We will incorporate these corrections in the revised manuscript.

References

Clark, M. P., A. G. Slater, D. E. Rupp, R. A. Woods, J. A. Vrugt, H. V. Gup ta, T. Wagener, and L. E. Hay.: Framework for Understanding Structural Errors (FUSE): A modular framework to diagnose differences between hydrological models,Water Resour. Res., 44, W00B02, doi:10.1029/2007WR006735, 2008.

Fenicia, F., D. Kavetski, and H. H. G. Savenije.: Elements of a flexible approach for conceptual hydrological modeling: 1. Motivation and theoretical development, Water Resources Research, 47(11), W11510, 10.1029/2010wr010174, 2011.

Gibbs, M., McInerney, D., Humphrey, G., Thyer, M., Maier, H., Dandy, G., and Kavetski, D.: State updating and calibration period selection to improve dynamic monthly streamflow forecasts for an environmental flow management application. Hydrology and Earth System Sciences Discussions, 22(1), 871-887, 2018.

Duke, C., Godel, M., Koch, L., Suter, J., and Ladher, R.: A study of the economic

impact of the services provided by the Bureau of Meteorology, London Economics, United Kingdom, 2016.

---

## Author Response (AR1)

**Response to Referee #1**

**General comment:** This review is for Manuscript ID: hess-2018-214, entitled Evaluating Residual Error Approaches for Post-processing Monthly and Seasonal Streamflow Forecasts, authored by Fitsum Woldemeskel and coauthors. With this manuscript the authors' aim is to evaluate different residual error models, including logarithmic (Log), Log-Sinh, and Box-Cox transformation schemes, for postprocessing monthly and seasonal streamflow forecasts. Overall, the postprocessed streamflow forecasts demonstrate skillful, reliable and sharper forecasts compared to the uncorrected forecasts. Furthermore, postprocessor employing the Box-Cox transformation scheme demonstrate the sharpest forecasts, without sacrificing skill and reliability. This manuscript is generally clear, however, it reads like a book chapter rather than a journal article. I believe the results and conclusions are of interest to the HESS community, as well as to the operational forecasters. Thus, this manuscript is worthy of publication if the issues below are addressed.

**Author response:** We thank the reviewer for the positive assessment of our manuscript as well as for their constructive comments and useful suggestions to improve the manuscript further. We are pleased that the reviewer found our manuscript suitable for the HESS research community and the community of operational forecasters. We provide specific responses to review comments as follows.

**Major Comments**

**Referee comment 1:** The introduction needs better organization. Consider removing the unnecessary details about the statistical modelling system and hybrid system (P4-5, L86-95), which are irrelevant in the context of dynamic modeling. The literature review can be focused on the usefulness of POAMA-2 in advancing seasonal hydrological forecasting.

**Author response:** We agree that statistical and hybrid systems are not directly relevant in the context of dynamic modelling. In the revised version, we have shortened the description of statistical and hybrid approaches to focus on the essentials and avoid excessive details (lines 71-75).

Referee comment 2: Make a separate subsection for the study area, dataset and hydrological model.

a) Study area: Provide general information on the hydroclimatic conditions, types of events across different seasons, basin size range, and reason for selecting the particular catchments.

**Author response:** Thank you for this suggestion. We have now added separate subsection for study area (section 3.1) with additional details about the catchments studied. As we are evaluating 300 catchments, it will be difficult to provide detailed site specific information, however, we provided summarised information highlighting those points suggested by the reviewer.

b) Dataset: Provide detail information on rainfall forecast dataset from POAMA-2, including forecast lead time, total number of ensemble members, and forecast initialization time and frequency. POAMA-2 information (P7, L189-194) should be integrated into the "Section 3.1 Data".

Author response: We have now provided additional information about rainfall forecast using POAMA-2 in subsection 3.2 as well as integrated lines 189-194 (older manuscript version) in subsection 3.2.

c) Hydrological model: I am concerned about the details of the rainfall-runoff model GR4J used for the study. It is necessary that you explain better the following aspects of the model: lumped conceptual model or physically based model, spatial resolution of the model, and the selected routing method. How often is the model initialized to make the forecast runs?

**Author response**: Thanks for raising this issue. In the revised manuscript we have included additional information in section 2.2.2 to clarify some details of the GR4J model used. Please refer to response to comment #3 below for some related information.

**Referee comment 3:** If the model is calibrated, then consider adding a subsection to discuss the simulation performance. You need to mention the calibrated parameters, model warm-up period, calibration period and validation period. The simulation performance can be discussed using correlation coefficient, percent bias and Nash-Sutcliffe efficiency between the observed and simulated streamflow.

**Author response:** We have included the details that the reviewer deemed necessary on how we calibrated the GR4J model parameters. We use 5 years model warm-up during 1975-1979 as well as calibration and validation during 1980-2008 in a moving 5 years leave-out cross-validation scheme. We have now clarified these points in the revised manuscript in sections 2.2.2 and 3.4.

We considered adding a subsection to discuss simulation performance as suggested by the reviewer, however we choose not to do this, because (1) the paper is focussed on improving streamflow forecast performance as this is the operational goal of the Bureau of Meteorology. (2) streamflow forecast performance in a dynamic modelling system is a function of the combined effects of the rainfall forecasts and the rainfall-runoff model, and this is clearly captured in the performance evaluation of the "uncorrected" streamflow forecasts (3) this paper is not about attributing whether the errors in streamflow forecasts are due to errors in the rainfall forecasts or the errors in the hydrological model, nor is it about comparing multiple hydrological models to determine which is the best to be used for forecasting purposes. These are both valuable research topic, but outside the scope of this paper. (4) this paper is focussed on once we have the "uncorrected" streamflow forecasts, what is best residual error modelling approach to post-process this streamflow forecasts. (5) the paper is already quite long, with 11 Figures and 1 table – adding the simulation performance of a hydrological model for ~300 catchments would require another at least another 1-2 Figures while providing little value for the reasons outlined above. In the discussion (Section 5.1) we have added some discussion on the value of trialling alternative hydrological models as part of future research.

**Referee comment 4:** In order to support the operational forecasting system, the conclusions drawn here should be valid in the context of extreme events. Does the conclusions apply to flood events? For this, verification metrics can be computed by considering the flow amounts greater than that implied by a non-exceedance probability, in the sampled climatological probability distribution, of 0.95.

**Author response:** While seasonal streamflow forecasts have limited application for flood prediction purposes, the question is relevant for predicting drought events, where the seasonal forecasts have significant value. In this study we evaluated forecast performance separately for high and low flow months, which provides an indication of predictive ability for below-average flows (i. e., drought events). In addition, the results and conclusion regarding the best performing error model scheme and its performance apply for the extreme events. Evaluation of forecast performance for extreme events (e.g.

<5% of historical data) is challenging because we may only have very small sample, which will make it difficult to draw definitive conclusions. We have now included the following paragraph in the discussion (section 5.5; lines 630-640) to acknowledge this issue and recommend it for future investigations.

Streamflow forecasts thus provide crucial information to water managers and users regarding the future availability of water, thus helping reduce uncertainty in decision making. This information is particularly valuable to support decision during drought events. In this study, forecast performance is evaluated separately for high and low flow months – providing a clearer indication of predictive ability for flows that are above and below average, respectively. A detailed evaluation of forecasts for more extreme drought events is challenging as these events are correspondingly rarer. Limited sample size makes it difficult to make conclusive statements: e.g. if we focus on the lowest 5% of historical data with a 30 year record, we may only have roughly 1.5 samples for each month/season. The uncertainty arising from limited sample size requires further development of forecast verification techniques, potentially adapting some of the approaches used by Hodgkins et al. (2017).

**Referee comment 5:** Considering an operational forecasting situation, how feasible is it to run 166 ensemble members using 40 GR4J parameters, and produce 6640 daily streamflow forecasts?

**Author response:** Yes, it is feasible to run 166 ensemble members with 40 GR4J parameters, and the Bureau of Meteorology has been running such a system operationally for a few years now. Producing 6640 forecasts this way is important to maintain reliability of forecasts. The largest computational expense results from calibrating hydrological models and cross-validation exercise rather than updating streamflow forecasts once every month using 166 ensembles members. However, the calibration and cross-validation exercise is typically done using a single observed rainfall timeseries. We also use high performance computing (HPC) facilities available at the Bureau of Meteorology and the National Computing Infrastructure (NCI) for calibrating hydrological models, which significantly reduces overall computation time. We have highlighted this in the revised manuscript in lines 181-182 and 190-193.

**Referee comment 6:** In the context of seasonal forecasting, different studies have demonstrated the combined ability of preprocessing meteorological forcing and postprocessing streamflow forecast to produce better streamflow forecasts. However, the study here only implements postprocessing. Was the meteorological forcing preprocessed? If not the case, it could be a topic of discussion, as a recommendation for future work to investigate the performance of residual error models in the context of preprocessing and postprocessing.

**Author response:** We use the analogue approach to downscale gridded POAMA-2 GCM rainfall forecast to catchment scale forecast, which can be considered as a form of rainfall forecast preprocessing. We have highlighted this point in the revised manuscript (section 3.2, line 308).

**Minor Comments**

Referee comment 7: Figure 8: Mention the units in the Y-axis for streamflow.

**Author response**: Thank you for this suggestion. We have now included units in the Y-axis in Figure 8.

**Referee comment 8**: Figure 8: Is there any reason for selecting Dieckmans Bridge catchment as a representative site for the analysis. Why is the time series plotted only for the period of 2003-2007? Is this a random selection?

**Author response:** Dieckmans Bridge catchment is selected as it is reflective of the results and conclusions across all catchments. That is, applying BC0.2 at this catchment resulted in better sharpness compared to applying Log and Log-Sinh while maintaining comparable CRPSS and reliability for high and low flow months. This is shown in Figure 9. The period 2003-2007 in Figure 8 is chosen as this period shows the difference in the forecast interval between the raw and three error models more clearly. We have now highlighted this point in the revised manuscript (section 4.2, lines 480-483).

**Referee comment 9:** Figure 9a: Replace "CRPS" with "CRPSS" in the Y-axis. **Referee comment 10:** P8 L200-204: Integrate this paragraph into the introduction. **Referee comment 11:** P9 L233: Provide a reference to the statement: "the parameters are estimated based on the methods of moments." **Referee comment 12:** P13 L365: Define the variable "y" in Equation 11.

Referee comment 13: P13 L367: How do you define the Heaviside step function?

Referee comment 14: P16 L444: Fix the typo for "Figure 45i".

Referee comment 15: P18 L495: Replace "unprocessed" with" uncorrected".

Referee comment 16: P18 L501: Define the acronyms: "NSW", "QLD" and "NT" when used for the first time.

Author response: We thank the reviewer for pointing out the above editorial corrections (comments 9-16). We have now incorporated all these corrections in the revised manuscript.

**Referee comment 17:** It may be good idea to provide a standard name for the streamflow postprocessing technique implemented in the study, is it a new technique? If not, then provide a suitable reference to the postprocessing technique.

**Author response:** We thank the reviewer for this suggestion. The residual error model approach used in this study is not new (e.g. the Box-Cox / power transformation has been introduced by Box and Cox, 1964; see McInerney et al., 2017 for detailed analysis), however, the application of it for post-processing monthly and seasonal streamflow forecasting in national forecasting system is new. This is clear from the presentation of Sections 2.3 and 2.4 which cite previous work and from sentences such as on lines 131-132 where we note that we are checking if findings obtained in case studies on daily streamflow prediction using observed rainfall data hold in applications with seasonal streamflow prediction using forecast rainfall.

**References**

Box, G. E. P. and Cox, D. R.: An analysis of transformations, Journal of the Royal Statistical Society, Series B. 26 (2): 211–252. JSTOR 2984418. MR 0192611.

Hodgkins, G. A., P. H. Whitfield, D. H. Burn, J. Hannaford, B. Renard, K. Stahl, A. K. Fleig, H. Madsen, L. Mediero, J. Korhonen, C. Murphy, and D. Wilson.: Climate-driven variability in the occurrence

of major floods across North America and Europe, J Hydrol, 552, 704-717, 10.1016/j.jhydrol.2017.07.027.

McInerney, D., Thyer, M., Kavetski, D., Lerat, J. and Kuczera, G.: Improving probabilistic prediction of daily streamflow by identifying Pareto optimal approaches for modeling heteroscedastic residual errors, Water Resour. Res., 53(3), 2199–2239, doi:10.1002/2016WR019168, 2017.

**Response to Referee #2**

General comment: This paper presents a comparison of three variants of a post-processing approach for long-range (here monthly to seasonal) streamflow forecasts in Australia. The paper is well-written and easy to read. The research is interesting for several reasons. First, the topic of long-range forecasting and especially how the skill of such forecasts can be improved is currently raising a lot of attention from hydrologists. There are many practical applications for which seasonal forecasts are required for decision-making. Second, Australia is a vast country which includes a broad range of hydroclimatic conditions, and the authors efficiently gathered a data base of 300 catchments, to ensure that their findings are as generalizable as possible. The authors explicitly address the specific case of dry catchments and low-flow periods, which is of great practical interest in several areas on the planet. Indirectly, the research has implications for socio-economic issues, as the management of water-scarce catchments could benefit from better seasonal forecasts. The paper also fits well within the scope of HESS.

This paper is definitely suitable for publication in HESS. However, I do have a few specific comments and suggestions for the authors to improve it prior to publication. In my opinion, all comments are minor.

**Author response:** We thank the reviewer for this very positive assessment of our manuscript and for recognising its practical importance. We are pleased that reviewer found our manuscript suitable for publication in HESS. We provide response to the review comments as follows.

**Specific comments/suggestions/guestions:**

Referee comment 1: Data assimilation/link with short-term forecasts

From the paper, I am not entirely sure how GR4J's state variables are managed (see also my next comment). From what I understand, there is no data assimilation at all. Perhaps this can be justified in the context of long-range forecasts as the effect of data assimilation would fade out quickly (probably before the one month horizon).

I was also wondering if there is a link (operationaly at BoM) between short- and longrange forecasts. Surely the hydrological model is the same, but what about the meteorological forecasts? Are the shortand long-ranges connected in some way? Surely, in operations, there must be a certain form of data assimilation for short lead times.

Considering the above interrogations, I would very much appreciate short comment regarding data assimilation in the paper.

**Author response:** The reviewer is correct that we have not used data assimilation to update GR4J state variables. However, data assimilation of ocean observations has been incorporated in the climate model (POAMA2.0) from which the rainfall forecasts have been obtained. We agree with the reviewer that the benefit of data assimilation for seasonal forecasts is limited. However, Gibbs et al. (2018) showed that monthly streamflow forecasting could benefit from state updating when the issue of non-stationarity is also handled. This is something to be investigated further in the future. We have now highlighted this issue in the manuscript (section 5.6).

Regarding the link between short- and long range forecasts provided by the Bureau of Meteorology, the two have independent systems due to the different needs of the forecasting service stakeholders. Short-range forecasts require a daily update with a focus on timely delivery of forecasts to anticipate quickly evolving events. Long-range forecast are more connected to longer-term decision making, which requires monthly update and statistically reliable forecasts. In addition, the GCM inputs are sourced from different models: short-term streamflow forecasts use weather forecasting model with limited modelling of ocean dynamic, whereas long range streamflow forecasts use climate outlook model with strong coupling between ocean and atmospheric models. As of now, updating state

variables through data assimilation is also not yet implemented in short-range forecasts, but there are plans to incorporate this in the future.

Overall, the streamflow forecast relies on data assimilation included in the climate model and of robust hydrologic modelling technique highlighted in the paper. It is also worth to mention that the Bureau of Meteorology prioritised investments in developing hydrologic modelling within a robust uncertainty framework, followed by streamflow and rainfall post-processing. In our view, the incremental benefits from data assimilation is likely to be less than these components.

Referee comment 2: Simulation and Forecast steps vs model calibration and warm-up

Section 2.2 and 2.3: I am slightly puzzled by that division into "simulation step", which includes model calibration, and "forecast step". Reading the description of the "simulation step", one could thing that you re-calibrate the model several times, once before each forecasting step. Is that so? If so, why? You want the model parameters to be dynamic?

I would tend to think that what the steps would rather be (1) calibrate the model (40 times using the MCMC-based method you mention) once and for all, (2) simulate streamflow over the entire period and save the state variables (I assume no data assimilation) and (3) launch GR4J in "forecast mode", by fetching the appropriate state variables for a specific date and feeding the model with meteorological forecasts.

I would very much appreciate if you could clarify those issues in the paper. In particular, think that calibration should be separated from simulation.

**Author response:** The three steps the reviewer described are correct. However, there is an additional process in step 1, i.e., we estimate parameters in a moving 5 years leave-out cross-validation approach. This is done in order to validate forecasts with an observed data set independent from the dataset used for calibration. We do not re-calibrate the hydrological model prior to each forecast.

We use data from 1980-2008 for cross-validation with a model warm-up period of 5 years (i.e. 1975-1979).

We clarified these points in the paper as well as made the distinction between calibration and forecast clearer in section 2.2. We also briefly mentioned the simulation step in line 184.

Referee comment 3: Discussing the choice of model for dry catchments

Section 5, lines 535-536, you mention that "This finding can be attributed to the challenge of capturing key physical processes in modeling dry and ephemeral catchments (:::)". In my opinion, this sentence leads to questioning whether of not GR4J is an appropriate model for very dry catchments. I know this model very well and I can appreciate its many qualities. GR4J works well for a very wide variety of hydro-climatic conditions. In addition, I do understand the practicality of having only one (very simple) model for all catchments on the entire country. However, there is no soil per se in GR4J. It is a very simple conceptual model which cannot, for instance, model soil sealing phenomena for dry catchments. I don't see how this model could ever capture the physical processes, as mentioned in your sentence.

In my opinion, this issue (the choice of a very simple conceptual model) should be briefly discussed following lines 535-536.

**Author response:** We thank the reviewer for pointing out this issue. The reviewer is correct that the model structure of GR4J, in particular its simplifying assumptions, might be responsible for the relatively lower forecast skill in dry catchments as compared to wet. Another potential source of poor performance is the errors in the rainfall forecasts, because these dry catchments have so few rainfall events with a high frequency of convective events, which are challenging to forecast for the POAMA GCM with a 250km grid size. Our general experience is that uncertainty of rainfall forecast is typically far larger than the hydrologic uncertainty. Our intent with respect to hydrological modelling is to use a model that can perform as best as possible in different hydro-climatologic conditions without necessarily being

complicated and non-parsimonious, and GR4J has shown to perform well under a wide-range of hydroclimatic conditions (Perrin et al., 2003; Tuteja et al., 2011).

Whilst using a single simple conceptual model is attractive for a practical operational system, there may be gains in exploring alternative model structures for difficult catchments (e.g. Clark et al., 2008; Fenicia et al., 2011). We intend to explore such alternative model structures for difficult ephemeral catchments. We have now highlighted these issues in section 5.1 (lines 549-554).

It is also worth mentioning that forecasting in dry catchments will remain an issue regardless of the hydrological model used due to the limited amount of information contained in streamflow records (high number of zero flow values) and high frequency of convective storms.

**Referee comment 4: Citing papers from HESS Discussion**

In my opinion, citing papers from HESS Discussion should be discouraged. After all, there is no real filtering of the papers before they can be published in discussion. The revision process takes place around the Discussion paper. To me, a paper that never makes it to HESS (after the Discussion) should be considered as rejected, even though it remains publicly accessible on the web. You wouldn't cite a paper that was rejected from other "more traditional" journals for which the revision is not as public as for HESS. Of course you could argue that if a paper in Discussion receives excellent comments but never makes it to HESS, it could be a case where the authors purposefully decided not to spend time editing it according to the reviewer's comments and re-submitting it. In my opinion, this practice, if it exists, should not be encouraged. Again, it wouldn't be possible with the majority of other journals.

Therefore, I would very strongly recommend that you remove all references to HESS Discussion. Set (2006) should therefore not be cited.

The citation for Mendoza et al (2017) should be updated as it is now published. Same for Turner et al (2017). The titles have also changed in the published version.

Author response: We agree with the suggestion not to cite HESS Discussion papers. Therefore, we will remove or modify the above references as appropriate in the revised manuscript. We will also update the references as suggested.

**Referee comment 5: Forecasts' value**

Section 5.3 lines 584-587, you briefly touch on the issue of forecasts value. I personally don't think measures of skill could ever be linked to the socio-economic value of forecasts. Most studies focussing on forecast values in the current literature largely over-simplify the problem. For the issue of forecasts value to be tackles in a more realistic way, researchers from humanities and social sciences would inevitably have to be involved. Forecasts value involves complex issues related to human psychology, economic theory, communication, social studies, etc. See for instance Morss et al. (2010), Matte et al. (2017), Toon et al. (2017) and Solin et al (2018).

In my opinion, forecasts skill is a pre-requisite for forecast value but in no way a guaranty. I don't see how metrics related strictly to the skill of a forecast (as in comparing the forecast to observation) could be a predictor of forecasts value on their own.

Morss et al (2017) Examining the use of weather forecasts in decision scenarios: results from a US survey with implications for uncertainty communication, METEOROLOGICAL APPLICATIONS, 17(2), 149-162

Matte et al (2017) Moving beyond the cost-loss ratio: economic assessment of streamflow forecasts for a risk-averse decision maker, HYDROLOGY AND EARTH SYSTEM SCIENCES, 21 (6), 2967-2986.

Toon et al (2017) Integrating Household Risk Mitigation Behavior in Flood Risk Analysis: An Agent-Based Model Approach, RISK ANALYSIS, 37 (10), 1977-1992

Solin et al (2018) Vulnerability assessment of households and its possible reflection in flood risk management: The case of the upper Myjava basin, Slovakia, INTERNATIONAL JOURNAL OF DISASTER RISK REDUCTION, 28, 640-652.

**Author response:** We thank the reviewer for these insights on the value of forecasts as well as for the suggestion of relevant literatures. We agree with the reviewer that forecast skill is a pre-requisite but not a guarantee of its value. A link between skill and value is a very complex issue as mentioned by the reviewer. We have now highlighted this issue in section 5.3 (lines 603-608) and cited some of the above references. In this regard, the Bureau actively works with its stakeholders to provide evidence about forecast value by developing application case studies (http://www.bom.gov.au/water/ssf/case\_studies.shtml). A recent socio-economic study conducted by London Economics has also highlighted the value of seasonal forecasts (Duke et al. 2016).

Referee comment 6: Typos/spelling/format/figures

- Page 10 line 255: I think the word " trial" should be replaced by "tried".

- Page 13 equation 11: The CRPS is usually computed by averaging the values over a large sample of forecasts-observation groups. Therefore, I think it is important that equation (11) be modified to be more explicit about this averaging.

- Page 14 line 388: "lead to misleading" is a bit strange to read. I would advise rephrasing

- Page 15 lines 413-414: there seem to be an awkward space between those two lines.

Please verify.

- Page 16 lines 443-444: Is "from in excess of 150%" the correct phrasing? Also, there is a typo in the parenthesis "(Figure 45i)".

- Page 18 line 493: remove comma after "scheme"

- Page 37, figure 8: please include the units for streamflow (y axis) on this figure. In addition, I am not entirely sure I understand the time step (x axis). Counting the points, I understand that the time step is one month, which would be coherent with the text, but not explicitly specified for this figure. In my opinion the x axis label could also be clearer.

- Page 38 figure 9: An "S" is missing for the y axis label of the top row. It should be CRPSS and not CRPS.

**Author response**: We thank the reviewer for pointing out the above editorial corrections. We have now incorporated all of these corrections in the revised manuscript.

**References**

- Clark, M. P., A. G. Slater, D. E. Rupp, R. A. Woods, J. A. Vrugt, H. V. Gupta, T. Wagener, and L. E. Hay.: Framework for Understanding Structural Errors (FUSE): A modular framework to diagnose differences between hydrological models, Water Resour. Res., 44, W00B02, doi:10.1029/2007WR006735, 2008.
- Duke, C., Godel, M., Koch, L., Suter, J., and Ladher, R.: A study of the economic impact of the services provided by the Bureau of Meteorology, London Economics, United Kingdom, 2016.
- Fenicia, F., D. Kavetski, and H. H. G. Savenije.: Elements of a flexible approach for conceptual hydrological modeling: 1. Motivation and theoretical development, Water Resources Research, 47(11), W11510, 10.1029/2010wr010174, 2011
- Gibbs, M., McInerney, D., Humphrey, G., Thyer, M., Maier, H., Dandy, G., and Kavetski, D.: State updating and calibration period selection to improve dynamic monthly streamflow forecasts for an environmental flow management application. Hydrology and Earth System Sciences Discussions, 22(1), 871-887, 2018.
- Perrin, C., Michel, C. and Andréassian, V.: Improvement of a parsimonious model for streamflow simulation, J. Hydrol., 279(1–4), 275–289, doi:10.1016/S0022-1694(03)00225-7, 2003.

Tuteja, N. K., Zhou, S., Lerat, J., Wang, Q. J., Shin, D. and Robertson, D. E.: Overview of Communication Strategies for Uncertainty in Hydrological Forecasting in Australia, in Handbook of Hydrometeorological Ensemble Forecasting, edited by Q. Duan, F. Pappenberger, J. Thielen, A. Wood, H. L. Cloke, and J. C. Schaake, pp. 1–19, Springer Berlin Heidelberg, Berlin, Heidelberg., 2016.

| Evaluating post-processing approaches for monthly and seasonal streamflow forecasts       Deleted:         Fitsum Woldemeskel (1) , David Meinerney (2) , Julien Leraf (3) , Mark Thyer (3) , Dmitri Kavetski (1,4) , Deleted:       Deleted:         (1) Bureau of Meteorology, VIC, Australia       Deleted:         (2) School of Civil, Environmental and Mining Engineering, University of Adelaide, SA, Australia       Hereorology, ACT, Australia         (3) Bureau of Meteorology, ACT, Australia       Hereorology, ACT, Australia         (4) School of Engineering, University of Newcastle, Callaghan, NSW, Australia       Hereorology, ACT, Australia         (5) Correspondence email: fitsum.woldemeskel@bom.gov.au       Deleted: :         (1)       Engineering, University of Newcastle, Callaghan, NSW, Australia       Deleted: :         (2)       Engineering, University of Newcastle, Callaghan, NSW, Australia       Engineering, University of Newcastle, Callaghan, NSW, Australia         (2)       Engineering, University of Newcastle, Callaghan, NSW, Australia       Engineering, University of Newcastle, Callaghan, NSW, Australia         (2)       Engineering, University of Newcastle, Callaghan, NSW, Australia       Engineering, University of Newcastle, Callaghan, NSW, Australia         (3)       Engineering, University of Newcastle, Callaghan, NSW, Australia       Engineering, University of Newcastle, Callaghan, NSW, Australia         (2)       Engineering, University of Newcastle, Callaghan, NSW, Australia       Engineerin                                                                                                                                                                                                                                                                                                                                              |                                                                                                                                                                                                                                                                  |                         |
|--------------------------------------------------------------------------------------------------------------------------------------------------------------------------------------------------------------------------------------------------------------------------------------------------------------------------------------------------------------------------------------------------------------------------------------------------------------------------------------------------------------------------------------------------------------------------------------------------------------------------------------------------------------------------------------------------------------------------------------------------------------------------------------------------------------------------------------------------------------------------------------------------------------------------------------------------------------------------------------------------------------------------------------------------------------------------------------------------------------------------------------------------------------------------------------------------------------------------------------------------------------------------------------------------------------------------------------------------------------------------------------------------------------------------------------------------------------------------------------------------------------------------------------------------------------------------------------------------------------------------------------------------------------------------------------------------------------------------------------------------------------------------------------------------------------------------------------------------------------------------------------------------------------------------------------------------------------------------------------------------|------------------------------------------------------------------------------------------------------------------------------------------------------------------------------------------------------------------------------------------------------------------|-------------------------|
| Evaluating post-processing approaches for monthly and seasonal streamflow forecasts Fitsum Woldeneskel (1) , David McInerney (2) , Julien Lerat (3) , Mark Thyer (2) , Dmitri Kavetski (2,4) , Deleted: Daehyok Shin (1) , Narendra Tutigia (3) and George Kuczera (4) (1) Bureau of Meteorology, VIC, Australia (2) School of Civil, Environmental and Mining Engineering, University of Adelaide, SA, Australia (3) Bureau of Meteorology, ACT, Australia (4) School of Engineering, University of Newcastle, Callaghan, NSW, Australia Correspondence email: fitsum.woldemeskel@bom.gov.au                                                                                                                                                                                                                                                                                                                                                                                                                                                                                                                                                                                                                                                                                                                                                                                                                                                                                                                                                                                                                                                                                                                                                                                                                                                                                                            |                                                                                                                                                                                                                                                                  |                         |
| streamflow forecasts Fitsum Woldemeskel (1) , David McInerney (2) , Julien Lerat (3) , Mark Thyer (2) , Dmitri Kavetski (2,4) , Deleted: Daehyok Shin (1) , Narendra Tuteja (3) and George Kuczera (4) (1) Bureau of Meteorology, VIC, Australia (2) School of Civil, Environmental and Mining Engineering, University of Adelaide, SA, Australia (3) Bureau of Meteorology, ACT, Australia (4) School of Engineering, University of Neweastle, Callaghan, NSW, Australia Correspondence email: fitsum.woldemeskel@bom.gov.au                                                                                                                                                                                                                                                                                                                                                                                                                                                                                                                                                                                                                                                                                                                                                                                                                                                                                                                                                                                                                                                                                                                                                                                                                                                                                                                                                                            | Evaluating post-processing approaches for monthly and seasonal                                                                                                                                                                                                   |                         |
| Fitsum Woldemeskel (1) , David McInerney (2) , Julien Lerat (3) , Mark Thyer (3) , Dmitri Kavetski (2,4) ,       Deleted:         Daehyok Shin (1) , Narendra Tuteja (2) and George Kuczera (4) (1) Bureau of Meteorology, VIC, Australia         (2) School of Civil, Environmental and Mining Engineering, University of Adelaide, SA, Australia       (3) Bureau of Meteorology, ACT, Australia         (4) School of Engineering, University of Newcastle, Callaghan, NSW, Australia       Correspondence email: fitsum.woldemeskel@bom.gov.au         Correspondence email: fitsum.woldemeskel@bom.gov.au       Deleted:         Image: State of the st | streamflow forecasts                                                                                                                                                                                                                                             |                         |
| (1) Bureau of Meteorology, VIC, Australia (2) School of Civil, Environmental and Mining Engineering, University of Adelaide, SA, Australia (3) Bureau of Meteorology, ACT, Australia (4) School of Engineering, University of Newcastle, Callaghan, NSW, Australia Correspondence email: fitsum.woldemeskel@bom.gov.au Peleted: 1 • • • • • • • • • • • • • • • • • • •                                                                                                                                                                                                                                                                                                                                                                                                                                                                                                                                                                                                                                                                                                                                                                                                                                                                                                                                                                                                                                                                                                                                                                                                                                                                                                                                                                                                                                                                                                                                                                                                                   | Fitsum Woldemeskel (1) , David McInerney (2) , Julien Lerat (3) , Mark Thyer (2) , Dmitri Kavetski (2,4) , Daehyok Shin (1) , Narendra Tuteja (3) and George Kuczera (4) | Deleted:                |
| (2) School of Civil, Environmental and Mining Engineering, University of Adelaide, SA, Australia
(3) Bureau of Meteorology, ACT, Australia
(4) School of Engineering, University of Newcastle, Callaghan, NSW, Australia
Correspondence email: fitsum.woldemeskel@bom.gov.au                                                                                                                                                                                                                                                                                                                                                                                                                                                                                                                                                                                                                                                                                                                                                                                                                                                                                                                                                                                                                                                                                                                                                                                                                                                                                                                                                                                                                                                                                                                                                                                                                                                                                                            | (1) Bureau of Meteorology, VIC, Australia                                                                                                                                                                                                                        |                         |
| (3) Bureau of Meteorology, ACT, Australia
(4) School of Engineering, University of Newcastle, Callaghan, NSW, Australia
Correspondence email: fitsum.woldemeskel@bom.gov.au                                                                                                                                                                                                                                                                                                                                                                                                                                                                                                                                                                                                                                                                                                                                                                                                                                                                                                                                                                                                                                                                                                                                                                                                                                                                                                                                                                                                                                                                                                                                                                                                                                                                                                                                                                                                                | (2) School of Civil, Environmental and Mining Engineering, University of Adelaide, SA, Australia                                                                                                                                                                 |                         |
| (4) School of Engineering, University of Newcastle, Callaghan, NSW, Australia
Correspondence email: fitsum.woldemeskel@bom.gov.au                                                                                                                                                                                                                                                                                                                                                                                                                                                                                                                                                                                                                                                                                                                                                                                                                                                                                                                                                                                                                                                                                                                                                                                                                                                                                                                                                                                                                                                                                                                                                                                                                                                                                                                                                                                                                                                             | (3) Bureau of Meteorology, ACT, Australia                                                                                                                                                                                                                        |                         |
| Correspondence email: fitsum.woldemeskel@bom.gov.au                                                                                                                                                                                                                                                                                                                                                                                                                                                                                                                                                                                                                                                                                                                                                                                                                                                                                                                                                                                                                                                                                                                                                                                                                                                                                                                                                                                                                                                                                                                                                                                                                                                                                                                                                                                                                                                                                                                                              | (4) School of Engineering, University of Newcastle, Callaghan, NSW, Australia                                                                                                                                                                                    |                         |
| V V Poleted: 1 1 1  Formatted: Font: Bold Formatted: Normal, Left                                                                                                                                                                                                                                                                                                                                                                                                                                                                                                                                                                                                                                                                                                                                                                                                                                                                                                                                                                                                                                                                                                                                                                                                                                                                                                                                                                                                                                                                                                                                                                                                                                                                                                                                                                                                                                                                                                                                |                                                                                                                                                                                                                                                                  |                         |
| Formatted: Font: Bold Formatted: Normal, Left                                                                                                                                                                                                                                                                                                                                                                                                                                                                                                                                                                                                                                                                                                                                                                                                                                                                                                                                                                                                                                                                                                                                                                                                                                                                                                                                                                                                                                                                                                                                                                                                                                                                                                                                                                                                                                                                                                                                                    | Υ                                                                                                                                                                                                                                                                | Deleted: 1              |
| Formatted: Font: Bold Formatted: Normal, Left                                                                                                                                                                                                                                                                                                                                                                                                                                                                                                                                                                                                                                                                                                                                                                                                                                                                                                                                                                                                                                                                                                                                                                                                                                                                                                                                                                                                                                                                                                                                                                                                                                                                                                                                                                                                                                                                                                                                                    |                                                                                                                                                                                                                                                                  | 1                       |
| Formatted: Normal, Left                                                                                                                                                                                                                                                                                                                                                                                                                                                                                                                                                                                                                                                                                                                                                                                                                                                                                                                                                                                                                                                                                                                                                                                                                                                                                                                                                                                                                                                                                                                                                                                                                                                                                                                                                                                                                                                                                                                                                                          |                                                                                                                                                                                                                                                                  | Formatted: Font: Bold   |
|                                                                                                                                                                                                                                                                                                                                                                                                                                                                                                                                                                                                                                                                                                                                                                                                                                                                                                                                                                                                                                                                                                                                                                                                                                                                                                                                                                                                                                                                                                                                                                                                                                                                                                                                                                                                                                                                                                                                                                                                  |                                                                                                                                                                                                                                                                  | Formatted: Normal, Left |
|                                                                                                                                                                                                                                                                                                                                                                                                                                                                                                                                                                                                                                                                                                                                                                                                                                                                                                                                                                                                                                                                                                                                                                                                                                                                                                                                                                                                                                                                                                                                                                                                                                                                                                                                                                                                                                                                                                                                                                                                  |                                                                                                                                                                                                                                                                  |                         |

**Abstract**

Streamflow forecasting is prone to substantial uncertainty due to errors in meteorological forecasts, hydrological model structure and parameterization, as well as in the observed rainfall and streamflow data used to calibrate the models. Statistical streamflow post-processing is an important technique available to improve the probabilistic properties of the forecasts. This study evaluates post-processing approaches based on three transformations - logarithmic (Log), log-sinh (Log-Sinh) and Box-Cox with  $\lambda = 0.2$  (BC0.2) – and identifies the best performing scheme for post-processing monthly and seasonal (3-months) streamflow forecasts, such as those produced by the Australian Bureau of Meteorology. Using the Bureau's operational dynamic streamflow forecasting system, we carry out comprehensive analysis of the three post-processing schemes across 300 Australian catchments with a wide range of hydro-climatic conditions. Forecast verification is assessed using reliability and sharpness metrics, as well as the Continuous Ranked Probability Skill Score (CRPSS). Results show that the uncorrected forecasts (i.e. without post-processing) are unreliable at half of the catchments. Post-processing of forecasts substantially improves reliability, with more than 90% of forecasts classified as reliable. In terms of sharpness, the BC0.2 scheme substantially outperforms the Log and Log-Sinh schemes. Overall, the BC0.2 scheme achieves reliable and sharper-than-climatology forecasts at a larger number of catchments than the Log and Log-Sinh transformations. The improvements in forecast reliability and sharpness achieved using the BC0.2 post-processing scheme will help water managers and users of the forecasting service to make better-informed decisions in planning and management of water resources.

Keywords: seasonal streamflow forecasts, post-processing, Box-Cox transformation

**Key points**

- Uncorrected and post-processed streamflow forecasts (using three transformations, namely Log, Log-Sinh and BC0.2) are evaluated over 300 diverse Australian catchments.
- Post-processing enhances streamflow forecast reliability, increasing the percentage of catchments with reliable predictions from 50% to over 90%.
- 3. The BC0.2 transformation achieves substantially better forecast sharpness than the Log-sinh and Log transformations, particularly in dry catchments.

| D | eleted: residual error models, based on the |  |
|---|---------------------------------------------|--|
| D | eleted: transformations respectively        |  |
| D | eleted:                                     |  |
|   |                                             |  |
| D | eleted: sites                               |  |
| D | eleted:                                     |  |
| D | eleted: significantly                       |  |
| D | eleted: .                                   |  |

**1 Introduction**

Hydrological forecasts provide crucial supporting information on a range of water resource management decisions, including (depending on the forecast lead-time) flood emergency response, water allocation for various uses, and drought risk management (Li et al., 2016; Turner et al., 2017). The forecasts, however, should be thoroughly verified and proved to be of sufficient quality to support decision-making and to meaningfully benefit the economy, environment and society.

Sub-seasonal and seasonal streamflow forecasting systems can be broadly classified as dynamic or statistical (Crochemore et al., 2016). In *dynamic* modelling systems, a hydrological model is usually developed at a daily time-step and calibrated against observed streamflow using historical rainfall and potential evaporation data. Rainfall forecasts from a numerical climate model are then used as an input to produce daily streamflow forecasts, which are then aggregated to the time scale of interest and post-processed using statistical models (e.g. Bennett et al., 2017; Schick et al., 2018). In *statistical* modelling systems a statistical model based on relevant predictors, such as antecedent rainfall and streamflow, is developed and applied directly at the time scale of interest (Robertson and Wang, 2009, 2011; Lü et al., 2016; Zhao et al., 2016), Hybrid systems that combine aspects of dynamic and statistical approaches have also been investigated (Humphrey et al., 2016; Robertson et al., 2013a),

Examples of operational services based on the dynamic approach include the Australian Bureau of Meteorology's dynamic modelling system (Laugesen et al., 2011; Tuteja et al., 2011; Lerat et al., 2015); the Hydrological Ensemble Forecast Service (HEFS) of the US National Weather Service (NWS) (Brown et al., 2014; Demargne et al., 2014); the Hydrological Outlook UK (HOUK) (Prudhomme et al., 2017); and the short-term forecasting European Flood Alert System (EFAS) (Cloke et al., 2013). Examples of operational services based on a statistical approach include the Bureau of Meteorology's Bayesian Joint Probability (BJP) forecasting system (Senlin et al., 2017).

Dynamic and statistical approaches have distinct advantages and limitations. Dynamic systems can potentially provide more realistic responses in unfamiliar climate situations, as it is possible to impose physical constraints in such situations (Wood and Schaake, 2008). In comparison, statistical models have the flexibility to include features that may lead to more reliable predictions. For example, the BJP model uses climate indices (e.g. NINO3.4), which are typically not used in dynamic approaches. That said, the suitability of statistical models for the analysis of non-stationary catchment and climate conditions is questionable (Wood and Schaake, 2008).

Streamflow forecasts built on hydrological models are affected by uncertainty in a number of factors, including rainfall forecasts, observed rainfall and streamflow data, as well as the parametric and structural uncertainty of the hydrological model. Progress has been made towards reducing biases and

Deleted: into Deleted: and Deleted: modelling systems Formatted: Font: Italic Deleted: commonly Deleted: to capture key hydrological processes. The model is Deleted: Once the model is calibrated, r Deleted: . Using rainfall forecast has been found to be beneficial flow forecasting Deleted: Formatted: Font: Italic Moved down [1]: Examples of operational services based on the dynamic approach include the Australian Bureau of Meteorology's dynamic modelling system (Laugesen et al., 2011; Tuteja et al., 2011; Lerat et al., 2013; the Hydrological Ensemble Forecast Service (HEFS) of the US National Weather Service (NWS) (Brown et al 2014; Demargne et al., 2014); the Hydrological Outlook UK et al., (HOUK) (Prudhomme et al., 2017); and the short-term forecasting European Flood Alert System (EFAS) (Cloke et al., 2013). Deleted: Withii. S Formatted: Font: Italic Formatted: Font: Not Italic Deleted: Here Field Code Changed Deleted: Wang et al., 2009; Deleted: Tang and Lettenmaier. 2010: Deleted: 
[revised manuscript text omitted]

| Deleted: This dynamic modelling system uses |  |
|---------------------------------------------|--|
| Deleted: d                                  |  |
| Deleted: as                                 |  |
| Deleted: s                                  |  |
| Deleted: In general, t                      |  |

| De                      | eleted: x 1 -                                      |  |  |
|-------------------------|---------------------------------------------------------------|--|--|
| Fo                      | rmatted: Font: Italic                                         |  |  |
| Fo                      | rmatted: Subscript                                            |  |  |
| De                      | eleted: x2-                                                   |  |  |
| Fo                      | rmatted: Font: Italic                                         |  |  |
| Fo                      | rmatted: Subscript                                            |  |  |
| Deleted: x3 -           |                                                               |  |  |
| De                      | eleted: x 4 -                                      |  |  |
| Fo                      | rmatted: Font: Italic                                         |  |  |
| Formatted: Subscript    |                                                               |  |  |
| Formatted: Font: Italic |                                                               |  |  |
| Fo                      | rmatted: Subscript                                            |  |  |
| De                      | Deleted: We use 5 years data (1975-1979) to warm-up the model |  |  |

and apply data from 1980-2008 for calibration in a moving 5 years leave-out cross-validation framework (see also section 3.3 for additional details). **Deleted:** We have not applied data assimilation technique to update

the GR4J state variables. This is partly due to limited effect of initial condition after a number of days resulting in minimal benefit as the benefit of data assimilation is minimal for the seasonal streamflow forecasting. However, Gibbs et al. (2018) showed that monthly streamflow forecasting could benefit from state updating in catchments which exhibited non-stationarity in rainfall-runoff response. Note that data assimilation of ocean observations has been implemented in the climate model (POAMA2.0) used for the rainfall forecast (Yin et al., 2011) (see Section 3.1.2 for additional details).

[revised manuscript text omitted]

$$Q_{t+1,j}^{PP} = Z^{-1}[Z(Q_{t+1}^F) + \eta_{t+1,j}]$$
(5)

Steps 1-4 are repeated for all ensemble members (6640 in our case).

Note that the above algorithm may occasionally generate negative streamflow predictions; such predictions are set to zero. This aspect is discussed in Section 5.6.

| λ | Deleted: , which is then |
|---|--------------------------|
| 1 | Formatted: English (US)  |
| ( | Deleted: 5.65.6          |
| - | Formatted: English (US)  |

| Deleted: a                                         |
|----------------------------------------------------|
|                                                    |
|                                                    |
| Deleted: $y_{t+1} = N(0, \sigma_y)$                |
| >                                                  |
| Deleted: (3b)                                      |
|                                                    |
| Deleted: $\mathcal{Y}_{t+1}$                       |
| Balanda Maria                                      |
| Deleted: ~ $N(0, \sigma_y)$                        |
| Deleted: assumed to follow a Gaussian distribution |
| Deleted: based on                                  |
| Deleted: set                                       |
| Deleted: to                                        |
| Deleted: set                                       |
| Deleted: to                                        |
|                                                    |
|                                                    |
| Deleted: or has been                               |
|                                                    |
| · · · · · · · · · · · · · · · · · · ·              |

| Deleted: | (note the additional subscript | j | for the ensemble |
|----------|--------------------------------|---|------------------|
| umber)   |                                |   |                  |

| De | leted: | determined |
|----|--------|------------|
|    |        |            |

r

(3)

**2.4 Transformations used in the post-processing model**

The observed streamflow and median streamflow forecast are transformed in Step 1 of streamflow postprocessing (Section 2.3.2), to account for the heteroscedasticity and skewness of the forecast residuals. We consider three transformations, namely the logarithmic, log-sinh and Box-Cox transformations.

**2.4.1 Logarithmic (Log) transformation**

The logarithmic (Log) transformation is

$$Z(Q) = \log(Q+c) \tag{6}$$

The offset *c* ensures the transformed flows are defined when Q = 0. Here we set  $c = 0.01 \times (\tilde{Q})_{ave}$ , where  $(\tilde{Q})_{ave}$  is the average observed streamflow over the calibration period. The use of a small fixed value for *c* is common in the literature for coping with zero flow events (Wang et al., 2012).

**2.4.2 Log-Sinh transformation**

The Log-Sinh transformation (Wang et al., 2012) is

$$Z(Q) = \frac{1}{b} \log[\sinh(a+bQ)]$$
(7)

The parameters *a* and *b* are calibrated for each month by maximising the p-value of the Shapiro-Wilk test (Shapiro and Wilk, 1965) for normality of the residuals,  $\nu$ . This pragmatic approach is part of the existing Bureau's operational dynamic streamflow forecasting system (Lerat et al., 2015).

**2.4.3 Box-Cox transformation**

The Box-Cox transformation (Box and Cox, 1964) is

$$Z(Q;\lambda,c) = \frac{(Q+c)^{\lambda} - 1}{\lambda}$$
(8)

where  $\lambda$  is a power parameter and  $c = 0.01 \times (\tilde{Q})_{ave}$ . Following the recommendations of McInerney et al. (2017), the parameter  $\lambda$  is fixed to 0.2.

**2.4.4 Rationale for selecting transformational approaches**

The Log transformation is a simple and widely used transformation; McInerney et al. (2017) reported that in daily scale modelling it produced the best reliability in perennial catchments (from a set of eight residual error schemes, including standard least squares, weighted least squares, BC, Log-Sinh and reciprocal transformation). However, the Log transformation performed poorly in ephemeral catchments, where its precision was far worse than in perennial ones,

| Deleted:
with doing | This avoids the need to calibrate $\lambda$ , and related problems so. |
|------------------------|------------------------------------------------------------------------|
|                        |                                                                        |
| Deleted:               | that is simple to implement                                            |
|                        |                                                                        |
|                        |                                                                        |

**Deleted: Deleted: 2.4.1 Deleted: To achieve Aim 2 of this study, w Deleted: different**

The Log-Sinh transformation is an alternative to the Log and BC transformations proposed by Wang et al. (2012) to improve the precision at higher flows. The Log-Sinh approach has been extensively applied to water forecasting problems (see for example, Del Giudice et al., 2013; Robertson et al., 2013b, Bennett et al., 2016). However, in daily scale streamflow modelling of perennial catchments, using observed rainfall, the Log-Sinh scheme did not improve on the Log transformation; its parameters tend to calibrate to values for which the Log-Sinh transformation effectively reduces to the Log transformation, (McInerney et al., 2017).

Finally, the BC transformation with fixed  $\lambda = 0.2$  is recommended by McInerney et al. (2017) as one of only two schemes (from the set of eight schemes listed earlier in this section) that achieve Pareto-optimal, performance in terms of reliability, precision and bias, across both perennial and ephemeral catchments. McInerney et al. (2017) also found that calibrating  $\lambda$  did not generally improve predictive performance, due to the inferred value being dominated by the fit to the low flows at the expense of the high flows.

**2.5 Summary of key terms**

In the remainder of the paper, the term "uncorrected forecasts" refers to streamflow forecasts obtained using steps in Section 2.2.3 and the term "post-processed forecasts" refers to forecasts based on a streamflow post-processing model, which includes the standardization and AR(1) model from Section 2.3 as well as a transformation (Log, Log-Sinh or BC0.2) from Section 2.4. As the post-processing schemes considered in this work differ solely in the transformation used, they will be referred to as the Log, Log-Sinh and BC0.2 schemes.

**3 Application**

**3.1 Study catchments**

The empirical case study is carried out over a comprehensive set of 300 catchments with locations shown in Figure 2. The figure also shows the Koppen climate zones. These catchments are selected as representative of the diverse hydro-climatic conditions across Australia. The catchment areas range from as small as 6 km2 to as Jarge as 23,2846 km2, with 90% of the catchments having areas below 6000 km2. The seasonal streamflow forecasting service of the Bureau of Meteorology is currently evaluating these 300 catchments as part of an expansion of their dynamic modelling system.

**3.2 Catchment data**

In each catchment, data from 1980-2008 is used. Observed daily rainfall data was obtained from the Australian Water Availability Project (AWAP) (Jeffrey et al., 2001). Potential evaporation and observed streamflow data were obtained from the Bureau of Meteorology.

[revised manuscript text omitted]

| vereteu:                 | verification                                                                                                                                                    |
|--------------------------|-----------------------------------------------------------------------------------------------------------------------------------------------------------------|
|                          |                                                                                                                                                                 |
| Delete                   | d: v                                                                                                                                                            |
|                          |                                                                                                                                                                 |
| maximise s
2005; Wilk | Dwn [3]: The goal of the forecasting exercise is to harpness without sacrificing reliability (Gneiting et al. is, 2011; Bourdin et al., 2014). Therefore |
| Deleted:                 | t                                                                                                                                                               |
| Deleted:                 | 3.4.3                                                                                                                                                           |
|                          |                                                                                                                                                                 |
| Deleted:                 | , RMSE and RMSEP results                                                                                                                                        |
| Deleted:                 | included                                                                                                                                                        |
| Deleted:                 | the current paper                                                                                                                                               |
|                          |                                                                                                                                                                 |
|                          |                                                                                                                                                                 |
|                          |                                                                                                                                                                 |
|                          |                                                                                                                                                                 |
| Deleted:                 | - "high flow" months are t                                                                                                                                      |
| Deleted:                 | , while                                                                                                                                                         |
| Deleted                  | are the 6 months with the lowest average streamflow                                                                                                             |
| veleted:                 |                                                                                                                                                                 |
| Deleted:                 | Note that although t                                                                                                                                            |

| Deleted:                                 |  |
|------------------------------------------|--|
|                                          |  |
| Deleted: s                               |  |
| Deleted: = 1, 2,, N, respectively |  |
| Deleted:                                 |  |
|                                          |  |
| Deleted: that are smaller                |  |
|                                          |  |
| Deleted: are wider                       |  |
| Deleted: consider                        |  |
| Deleted: consider                        |  |

i.e., the IQR at the 99 percentile, in order to detect forecasts with unreasonably long tails in their predictive distributions.

**3.5.3 CRPS skill score (CRPSS)**

The *CRPS* metric quantifies the difference between a forecast distribution and observations, as follows (Hersbach, 2000):

$$CRPS = \frac{1}{N} \times \sum_{i=1}^{N} \int_{-\infty}^{\infty} [F_i(y) - H_i\{y \ge y_o\}]^2 dy$$
(11)

where  $F_i$  is the cumulative distribution function (cdf) of the forecast for year,  $i_e y$  is the forecast variable (here streamflow) and  $y_o$  is the corresponding observed value.  $H_i\{y \ge y_o\}$  is the Heaviside step function, which equals  $\downarrow$  when the forecast values are greater than the observed value and equals 0 otherwise.

[revised manuscript text omitted]

| Del | leted: both                             |
|-----|-----------------------------------------|
| For | rmatted: Font: Italic                   |
| Del | leted: has a better                     |
| Del | leted: ness                             |
| Del | leted: Accordingly, a                   |
| Del | leted: (low)                            |
| Del | leted: (0-2 months)                     |
| Del | leted: reliable                         |
| Del | leted: that are shaper than climatology |
| Del | leted: we do not include the            |
| Del | leted: the CRPSS                        |
| Del | leted: provide                          |
| Del | leted: 3.4.3                            |
| Del | leted: In addition, t                   |
| Del | leted: The results are                  |
| Mo  | ved (insertion) [4]                     |
| Del | leted: (Sections 3.33.2 and 3.53.4).    |

[revised manuscript text omitted]

The findings for forecasts at the seasonal scale are as follows (Figure 11, and Table 1);

- Log scheme has the largest percentage (19%) of catchments with low summary skill and a relatively small percentage (9%) of catchments with high summary skill.
- Post-processing forecasts with the Log and Log-Sinh schemes reduces the percentages of catchments with low summary skill from 19% to 18% and 17% respectively. The percentage of catchments with high summary skill increases from 9% to 12% and 22% respectively.
- Post-processing with the BC0.2 scheme once again provides the best performance: it produces forecasts with low summary skill in only 2% of the catchments, and achieves high summary skill

| Deleted                            | Figure 11                                                                                                                                    |
|------------------------------------|----------------------------------------------------------------------------------------------------------------------------------------------|
| Deleted                            | : T                                                                                                                                          |
| Deleted                            | aggregates multiple verifications metrics; it                                                                                                |
| Deleted                            | exhibit a                                                                                                                                    |
| Deleted                            | : ness                                                                                                                                       |
| Deleted                            | that is better                                                                                                                               |
| Deleted
those with
and sharp | Catchments with high (low) summary skill are defined as
10-12 months (0-2 months) with forecasts that are reliabl
er than climatology. |
| Deleted                            | : At the                                                                                                                                     |
| Deleted                            | ; , we obtain the following key findings                                                                                                     |
|                                    |                                                                                                                                              |
| Delet                              | ed: ,                                                                                                                                        |
| Delet                              | ed:,                                                                                                                                         |

achieves nigh sun

in 54% of the catchments. As seen in Figure 11, similar to the case of monthly forecasts, the biggest improvements for seasonal forecasts occur in the NSW and Queensland regions of Australia.

Overall, Table 1 shows that, across all schemes, BC0.2 results in a larger percentage of catchments with low summary skill and a larger percentage of catchments with high summary skill. It can also be seen that the summary skills of post-processing approaches are lower for seasonal forecasts than for monthly forecasts.

**4.4 Summary of empirical findings**

Sections 4.1-4.3 show that post-processing achieves major improvements in reliability, as well as in CRPSS and sharpness, particularly in dry catchments. Although all three post-processing schemes under consideration provide improvements in some of the performance metrics, the BC0.2 scheme consistently produces better sharpness than the Log and Log-Sinh schemes, while maintaining similar reliability and CRPSS. This finding holds for both monthly and, to a less degree, seasonal forecasts. Of the three post-processing schemes, the BC0.2 scheme improves by the largest margin the percentage of catchments and the number of months where the post-processed forecasts are reliable and sharper than climatology.

**5 Discussion**

**5.1 Benefits of forecast post-processing**

A comparison of uncorrected and post-processed streamflow forecasts was provided in Section 4.1. Uncorrected forecasts have reasonable sharpness (except for in dry catchments), but suffer from low reliability: uncorrected forecasts are unreliable at approximately 50% of the catchments. In wet catchments, poor reliability is due to overconfident forecasts, which appears a common concern in dynamic forecasting approaches (Wood and Schaake, 2008). In dry catchments, uncorrected forecasts are both unreliable and exhibit poor sharpness. Post-processing is thus particularly important to correct for these shortcomings and improve forecast skill. In this study, all post-processing models provide a clear improvement in reliability and sharpness, especially in dry catchments. The value of postprocessing is more pronounced in dry catchments than in wet catchments (Figure 4, and Figure 5). This finding can be attributed to the challenge of capturing key physical processes in dry and ephemeral catchments (Ye et al., 1997), as well as the challenge of achieving accurate rainfall forecasts in arid areas. In addition, the simplifications inherent in any hydrological model, including the conceptual model GR4J used in this work, might also be responsible for the forecast skill being relatively lower in dry catchments than in wet catchments. Whilst using a single conceptual model is attractive for practical operational system, there may be gains in exploring alternative structures for ephemeral catchments (e.g. Clark et al., 2008; Fenicia et al., 2011). We intend to explore such alternative model structures for

| Deleted: Figure 11                                                                                                           |
|------------------------------------------------------------------------------------------------------------------------------|
| Deleted: shows that                                                                                                          |
|                                                                                                                              |
|                                                                                                                              |
|                                                                                                                              |
| Deleted: the summary skills of post-processing approaches are lower for seasonal forecasts than for monthly forecasts |

| Del | eted: produ  | ices        |      |  |  |
|-----|--------------|-------------|------|--|--|
| Del | eted: resid  | al error mo | dels |  |  |
|     |              |             |      |  |  |
|     |              |             |      |  |  |
|     |              |             |      |  |  |
| Del | eted: reside | al error mo | dels |  |  |
| Del | otod: sites  |             |      |  |  |

| Deleted: significant         |
|------------------------------|
| Deleted: Figure 4            |
| Deleted: modelling           |
|                              |
| Deleted: (REF).              |
| Deleted: y                   |
| Deleted: ng                  |
| Deleted: assumptions         |
| Deleted: of                  |
| Deleted: the conceptual GR4J |
| Deleted: relatively lower    |
| Deleted: as compared to      |

difficult ephemeral catchments. In such dry catchments, the hydrological model forecasts are particularly poor and leave a lot of room for improvement: post-processing can hence make a big difference on the quality of results.

**5.2 Interpretation of differences between post-processing schemes**

We now discuss the large differences in sharpness between the BC0.2 scheme versus the Log and Log-Sinh schemes. The Log-Sinh transformation was designed by Wang et al. (2012) to improve the reliability and sharpness of predictions, particularly for high flows, and has worked well as part of the statistical modelling system for operational streamflow forecasts by the Bureau of Meteorology. The Log-Sinh transformation has a variance stabilizing function that (for certain parameter values) tapers off for high flows. In theory, this feature can prevent the explosive growth of predictions for high flows that can occur with the Log and Box-Cox transformations (especially when  $\lambda < 0$ ).

McInerney et al. (2017) found that, when modelling perennial catchments at the daily scale, the Log-Sinh scheme did not achieve better sharpness than the Log scheme. Instead, the parameters for the Log scheme tended to converge to values for which the tapering off of the Log-Sinh transformation function occurs well outside the range of simulated flows, effectively reducing the Log-Sinh scheme to the Log scheme. In contrast, the Box-Cox transformation function with a fixed  $\lambda > 0$  gradually flattens as streamflow increases, and exhibits the "desired" tapering-off behaviour within the range of simulated flows. This behaviour leads to the Box-Cox scheme achieving, on average, more favourable variancestabilizing characteristics than the Log-Sinh scheme.

Our findings in this study confirm the insights of McInerney et al. (2017) – namely that the Log-Sinh scheme produces comparable sharpness to the Log scheme – across a larger number of catchments. This finding indicates that insights from modelling residual errors at the daily scale apply at least to some extent to streamflow forecast post-processing at the monthly and seasonal scales. Note the minor difference in the treatment of the offset parameter *c* in equation (6): in the Log scheme used in McInerney et al. (2017) this parameter is inferred, whereas in this study it is fixed a priori. This minor difference does not impact on the qualitative behaviour of the error models described earlier in this section. Overall, when used for post-processing seasonal and monthly forecasts in a dynamic modelling system, the BC0.2 scheme provides an opportunity to improve forecast performance further than is possible using the Log and Log-Sinh schemes.

**5.3 Importance of using multiple metrics to assess forecast performance**

The goal of the forecasting exercise is to maximise sharpness without sacrificing reliability (Gneiting et al., 2005; Wilks, 2011; Bourdin et al., 2014). The study results show that relying on a single metric for evaluating forecast performance can lead to sub-optimal conclusions. For example, if one considers the

| Deleted: resid | dual error model                     |
|----------------|--------------------------------------|
| Deleted: in or | rder                                 |
| Deleted: whe   | n used                               |
|                |                                      |
| Deleted: corr  | esponds                              |
| Deleted: to    |                                      |
|                |                                      |
| Deleted: 1     |                                      |
| Deleted: resid | dual error models                    |
|                |                                      |
|                |                                      |
| Deleted: ; i   |                                      |
| Deleted: 1     |                                      |
| Deleted: sche  | me                                   |
| Deleted: and   | hence                                |
| Deleted: effe  | ctively reduces                      |
| Deleted: erro  | r model                              |
| Deleted: whe   | n using                              |
| Deleted: has   | a variance-stabilizing function that |
| Deleted: i.e., | it                                   |
|                |                                      |

|   | Deleted: , as                                                                                            |
|---|----------------------------------------------------------------------------------------------------------|
| ( | Deleted: T                                                                                               |
|   | Deleted: further                                                                                         |
|   | Deleted: relative to what is currently                                                                   |
|   | Deleted: when used as residual error post-processor of forecasts in a dynamical modelling systems |

| Moved | (insertion) | [3] |
|-------|-------------|-----|
|       |             |     |

CRPSS metric alone, all post-processing schemes yield comparable performance and there is no basis for favouring any single one of them. However, once sharpness is taken into consideration explicitly, the BC0.2 scheme can be recommended due to substantially better sharpness than the Log and Log-Sinh schemes.

Similarly, comparisons based solely on CRPSS might suggest reasonable performance of the uncorrected forecasts: 55%-80% of months have CRPSS > 0 (with some variability across high/low flow months and monthly/seasonal forecasts. Yet once reliability is considered explicitly, it is found that uncorrected forecasts are unreliable at approximately 50% of the catchments. Note that performance metrics based on the CRPSS reflect an implicitly weighted combination of reliability, sharpness and bias characteristics of the forecasts (Hersbach, 2000). In contrast, the reliability and sharpness metrics are specifically designed to quantify reliability and sharpness attributes individually. These findings highlight the value of multiple independent performance metrics and diagnostics that evaluate specific (targeted) attributes of the forecasts, and highlight important limitations of aggregate measures of performance (Clark et al., 2011).

A number of challenges and questions remain in regards to selecting the performance verification metrics for specific forecasting systems and applications. An important question is how to include user needs into a forecast verification protocol. This could be accomplished by tailoring the evaluation metrics to the requirements of users. Another key question is to what extent do measures of forecast skill correlate to the economic and/or social value of the forecast? This challenging question was investigated by Murphy and Ehrendorfer (1987) and Wandishin and Brooks (2002), who found the relationship between quality and value of a forecast to be essentially nonlinear: an increase in forecast quality may not necessarily lead to a proportional increase in its value. This question requires further multi-disciplinary research, including human psychology, economic theory, communication and social studies (e.g. Matte et al., 2017; Morss et al., 2010).

**5.4 Jmportance of performance evaluation over large numbers of catchments**

When designing an operational forecast service for locations with streamflow regimes as diverse and variable as in Australia (Taschetto and England, 2009), it is essential to thoroughly evaluate multiple modelling methods over multiple locations to ensure the findings are sufficiently robust and general. This was the major reason for considering the large set of 300 catchments in our study. This setup also yields valuable insights into spatial patterns in forecast performance. For example, the Log and Log-Sinh schemes perform relatively well in catchments in South-Eastern Australia, and relatively worse in catchments in Northern and North-Eastern Australia (Figure 10 and Figure 11). In contrast, the BC0.2 scheme performs well across the majority of the catchments in all regions included in the evaluation. The evaluation over a large number of catchments in different hydro-climatic regions is clearly beneficial

| Deleted: (              |  |
|-------------------------|--|
| Deleted: depending      |  |
| Deleted: on             |  |
| Deleted: ), y           |  |
| Deleted: , for example, |  |
| Deleted: s              |  |
| Deleted: whereas        |  |
| Deleted: target         |  |
| Deleted: respectively   |  |

| -( | Deleted: Having said that this is a complex |
|----|---------------------------------------------|
| •( | Deleted: that spans                         |
| •( | Deleted: ple disciplines                    |
|    |                                             |

to establish the robustness of post-processing methods. Restricting the analysis to a smaller number of catchments would have led to less conclusive findings.

**5.5 Implication of results for water resource management**

The empirical results clearly show that the BC0.2 post-processing scheme improves forecast sharpness (precision) while maintaining forecast accuracy and reliability. As discussed below, this improvement in forecast quality offers an opportunity to improve operational planning and management of water resources.

The management of water resources, for example, deciding which water source to use for a particular purpose or allocating environmental flows, requires an understanding of the current and future availability of water. For water resources systems with long hydrological records, water managers have devised techniques to evaluate current water availability, water demand and losses. However, one of the main unknowns is the volume of future system inflows. Streamflow forecasts thus provide crucial information to water managers and users regarding the future availability of water, thus helping reduce uncertainty in decision making. This information is particularly valuable to support decision during drought events. In this study, forecast performance is evaluated separately for high and low flow months providing a clearer, indication of predictive ability for flows that are above and below average, respectively. A detailed evaluation of forecasts for more extreme drought events is challenging as these events are correspondingly rarer. Limited sample size makes it difficult to make conclusive statements; e.g. if we focus on the lowest 5% of historical data with a 30 year record, we may only have roughly 1.5 samples for each month/season, The uncertainty arising from limited sample size requires further development of forecast verification techniques, potentially adapting some of the approaches used by Hodgkins et al. (2017),

**5.6 Opportunities for further improvement in forecast performance,**

There are several opportunities to further improve the seasonal streamflow forecasting system. This section describes two such avenues, namely specialised treatment of zero flows and the use of data assimilation.

The post-processing approaches used in this work do not make special provision for zero flows in the observed data. Robust handling of zero flows in statistical models, especially in arid and semi-arid catchments, is an active research area (Wang and Robertson, 2011; Smith et al., 2015), and advances in this area are certainly relevant to seasonal streamflow forecasting.

The forecasting system used in this study does not implement state updating in the GR4J hydrological model, Gibbs et al. (2018) showed that monthly streamflow forecasting could benefit from state updating

| Deleted: we evaluated                                                                                                                                                                                                                                                                                                           |
|---------------------------------------------------------------------------------------------------------------------------------------------------------------------------------------------------------------------------------------------------------------------------------------------------------------------------------|
| Deleted: by undertaking a targeted evaluation on both                                                                                                                                                                                                                                                                           |
| Deleted: -                                                                                                                                                                                                                                                                                                                      |
| Deleted: the latter                                                                                                                                                                                                                                                                                                             |
| Deleted: n                                                                                                                                                                                                                                                                                                                      |
| Deleted: flow                                                                                                                                                                                                                                                                                                                   |
| Deleted: (                                                                                                                                                                                                                                                                                                                      |
| Deleted: )                                                                                                                                                                                                                                                                                                                      |
| Deleted: To handle                                                                                                                                                                                                                                                                                                              |
| Deleted: t                                                                                                                                                                                                                                                                                                                      |
| Deleted: created by this                                                                                                                                                                                                                                                                                                        |
| Deleted: the                                                                                                                                                                                                                                                                                                                    |
| Deleted: new                                                                                                                                                                                                                                                                                                                    |
| Deleted: for groups of catchments                                                                                                                                                                                                                                                                                               |
| Deleted: This will be undertaken as part of future research.                                                                                                                                                                                                                                                                    |
| Deleted: The results and conclusions clearly illustrate Tthe ability of the BC0.2 post-processing scheme to improve forecast sharpness (precision) while maintaining forecast accuracy and reliability and can hence this has the potential to lead to improved operational planning and management of water resources.¶ |

**Deleted: Treatment of zero flows**

in catchments which exhibited non-stationarity in rainfall-runoff response. Note that data assimilation of ocean observations has been implemented in the climate model (POAMA2) used for the rainfall forecast (Yin et al., 2011) (see Section 3.2 for additional details).

**6 Conclusions**

This study focused on developing robust streamflow forecast post-processing schemes for an operational forecasting service at the monthly and seasonal time scales. For such forecasts to be useful to water managers and decision-makers, they should be reliable and exhibit sharpness that is better than climatology.

We investigated streamflow forecast post-processing schemes based on residual error models employing three data transformations, namely the logarithmic (Log), log-sinh (Log-Sinh) and Box-Cox with  $\lambda = 0.2$ (BC0.2). The Australian Bureau of Meteorology's dynamic modelling system was used as the platform for the empirical analysis, which was carried out over 300 Australian catchments with diverse hydroclimatic conditions.

**The following empirical findings are obtained:**

- Uncorrected forecasts (no post-processing) perform poorly in terms of reliability, resulting in a mis-characterization of forecast uncertainties.
- All three post-processing schemes substantially improve the reliability of streamflow forecasts, both in terms of the dedicated reliability metric and in terms of the summary skill given by the CRPSS;
- 3. From the post-processing schemes considered in this work, the BC0.2 scheme is found best suited for operational application. The BC0.2 scheme provides the sharpest forecasts without sacrificing reliability, as measured by the reliability and CRPSS metrics. In particular, the BC0.2 scheme produces forecasts that are both reliable and sharper than climatology at substantially more catchments than the alternative Log and Log-Sinh schemes.

A major practical outcome of this study is the development of a robust streamflow forecast post- processing scheme that achieves forecasts that are consistently reliable and sharper than climatology. This scheme is well suited for operational application, and offers the opportunity to improve decision support, especially in catchments where climatology is presently used to guide operational decisions.

**7 Data availability**

The data underlying this research can be accessed from the following links: Observed rainfall data (http://www.bom.gov.au/climate/); POAMA rainfall forecast (http://poama.bom.gov.au/); and observed streamflow data (http://www.bom.gov.au/waterdata/).

[revised manuscript text omitted]

---

## Referee Report (RR1)

This review is for manuscript HESS-2018-214, entitled *Evaluating post-processing approaches for monthly and seasonal streamflow forecasts*, authored by Fitsum Woldemskel and coauthors. The paper is well written throughout, and I believe the results and conclusions are of interest to the HESS community. I found that the authors have addressed all of the comments from the previous reviews. Other than the following minor comment, I think the manuscript is ready for publication in HESS.

- In the case that the post-processed streamflow falls beyond the historical maxima/minima, how did you back transform it into the real space? Was the sensitivity of the different transformation schemes with the length of the calibration period investigated? If not the case, is there any suggestion for the length of the historical data requirement for effective implementation of different transformation schemes, mainly the BC0.2 scheme which is found best for operational application?

---

## Author Response (AR2)

**Response to Referee comments**

**General comment:** This review is for manuscript HESS-2018-214, entitled Evaluating post-processing approaches for monthly and seasonal streamflow forecasts, authored by Fitsum Woldemskel and coauthors. The paper is well written throughout, and I believe the results and conclusions are of interest to the HESS community. I found that the authors have addressed all of the comments from the previous reviews. Other than the following minor comment, I think the manuscript is ready for publication in

HESS.

***Author response**: We thank the reviewer for positive assessment of our manuscript and for finding*

*the paper of interest to the HESS community.*

**Specific comment 1:** In the case that the post-processed streamflow falls beyond the historical maxima/minima, how did you back transform it into the real space?

***Author response**: This is a good point worthy of clarification. We ensure the post-processed*

*streamflow forecasts are always positive (see Lines 223-224 and Lines 628-631) but do not apply an*

*upper limit, as is now explained on Lines 224-226 and Lines 632-633 of the revised manuscript.*

*In other words, we do not attempt to restrict the model from producing post-processed streamflow that*

*exceeds the historical maximum at the forecast site. This is somewhat similar to flood frequency*

*analysis, where a probability distribution is used to extrapolate beyond the historical maximum. There*

*is nothing technically wrong with extrapolating beyond the historical maximum, it simply depends on*

*the degree of confidence in the model. In this paper, the IQR ratio, used as part of the forecast*

*performance metrics, evaluates the range of the $99^{th}$ percentile – and is designed to detect*

*unreasonably long tails (i.e. extremes) in the predictive distributions (see Lines 633-635 in revised*

*manuscript). Hence, it goes some way towards evaluating the degree of confidence in high flow*

*forecasts.*

*We recognise that further research is needed to evaluate the realism of high flow forecasts and design*

*techniques for detecting and remedying such occurrences. This is now noted on Lines 636-637 of*

*revised manuscript.*

**Specific comment 2:** Was the sensitivity of the different transformation schemes with the length of the calibration period investigated? If not the case, is there any suggestion for the length of the historical data requirement for effective implementation of different transformation schemes, mainly the BC0.2 scheme which is found best for operational application?

*Author response: We have not investigated the sensitivity of the post-processing model (which includes the transformation scheme and the monthly parameters) to the length of calibration period. In this study we used a 29 year period (1980-2008) for calibration and evaluation (so estimation uncertainty is likely to be small), and have employed a cross-validation procedure to detect any loss in performance due to over-fitting. Therefore we are confident the conclusions are robust for data used in this study.*

*We agree that, if the calibration period is short, the uncertainty in the parameters of the post-processing model may be large. For example, if calibrating to 5 years of data only 5 data points would be available to calibrate monthly parameters. In such circumstances, parameter uncertainty analysis would be necessary, and the cross-validation would need to be re-done to detect any possible impacts.*

*These issues are now listed succinctly in Section 5.6 (Lines 638-644) as an opportunity to further understand and improve the post-processing model.*

[revised manuscript text omitted]